



# How frequent is natural cloud seeding over Switzerland?

Ulrike Proske[1], Verena Bessenbacher[1], Zane Dedekind[1], Ulrike Lohmann[1], and David Neubauer[1]

[1]Institute for Atmospheric and Climate Science, ETH Zürich, Zürich, Switzerland

**Correspondence:** Ulrike Proske (ulrike.proske@env.ethz.ch)

**Abstract.** Clouds and cloud feedbacks represent one of the largest uncertainties in climate projections. As the ice phase influences many key cloud properties and their lifetime, its formation needs to be better understood in order to improve climate and weather prediction models. Ice crystals sedimenting out of a cloud do not sublimate immediately but can survive certain distances and eventually fall into a cloud below. This natural cloud seeding can trigger glaciation and has been shown to

enhance precipitation formation. However, up to date an estimate of its occurrence frequency is lacking. In this study, we estimate the occurrence frequency of natural cloud seeding over Switzerland from satellite data and sublimation calculations.

    We use the DARDAR satellite product between April 2006 and October 2017 to estimate the occurrence frequency of multilayer cloud situations, where a cirrus cloud at $T < -35\,^\circ\mathrm{C}$ can provide seeds to a lower lying feeder cloud. These situations are found to occur in $31\,\%$ of the observations. Of these $42\,\%$ have a cirrus cloud above another cloud, separated, while in $58\,\%$ the

cirrus is part of a thicker cloud, with a potential for in-cloud seeding. Vertical distances between the cirrus and the lower-lying cloud are distributed uniformly between $100\,\mathrm{m}$ and $10\,\mathrm{km}$. They are found to not vary with topography. Seasonally, winter nights have the most multilayer cloud occurrences, in $38\,\%$ of the measurements. Additionally, in situ and liquid origin cirrus cloud size modes can be identified according to the ice crystal mean effective radius in the DARDAR data. Using sublimation calculations we show that in a significant number of cases the seeding ice crystals do not sublimate before reaching the lower

lying feeder cloud. Depending on whether bullet rosette, plate like or spherical crystals were assumed, $10\,\%$, $11\,\%$ or $20\,\%$ of the crystals, respectively, could provide seeds after sedimenting $2\,\mathrm{km}$.

    The high occurrence frequency of seeding situations and the survival of the ice crystals indicate that the seeder-feeder process and natural cloud seeding are widespread phenomena over Switzerland. This hints to a large potential for natural cloud seeding to influence cloud properties and thereby the Earth's radiative budget and water cycle, which should be studied globally.

Further investigations of the magnitude of the seeding ice crystals' effect on lower lying clouds are necessary to estimate the contribution of natural cloud seeding to precipitation.





## 1 Introduction

Clouds and cloud feedbacks contribute the largest uncertainty to projections of climate sensitivity in global climate models
(Cess et al., 1990; Soden and Held, 2006; Williams and Tselioudis, 2007; Boucher et al., 2013). Cloud microphysics, and especially cloud ice / water content, determine key cloud properties, such as their albedo and lifetime, and control precipitation formation (Mülmenstädt et al., 2015). The representation of the ice phase in clouds is therefore necessary to estimate the Earth's radiation budget and its response to climate change (Sun and Shine, 1995; Tan et al., 2016; Matus and L'Ecuyer, 2017; Lohmann and Neubauer, 2018) as well as to improve forecasts of precipitation in numerical weather prediction models. Natural
cloud seeding can be a source of ice crystals in clouds, lead to the glaciation of clouds and enhance precipitation. Moreover, the seeder-feeder mechanism has been associated with the enhancement of extreme precipitation and flooding (Rössler et al., 2014). An understanding of the seeder-feeder mechanism is therefore necessary to improve the representation of the cloud ice phase in weather and climate models, to improve weather forecasts of precipitation, and ultimately to reduce uncertainty in climate simulations.

The seeder-feeder mechanism was originally proposed to explain an observed enhancement of precipitation over mountains. In this classical setting, precipitation from an overlying "seeder" cloud falls into an orographic "feeder" cloud. In the lower cloud, the precipitation particles grow by accretion, coalescence, or riming, which leads to an enhancement of precipitation over the orography (Roe, 2005). This classical seeder-feeder mechanism has been observed in field studies in various locations (Dore et al., 1999; Purdy et al., 2005; Hill et al., 2007) and has been reproduced in a number of idealized modelling studies
(e.g. Carruthers and Choularton (1983); Robichaud and Austin (1988)).

Braham (1967) noted the possibility of ice crystals from cirrus clouds acting as seeds for ice formation in lower-lying warmer clouds. In this special case of the seeder-feeder mechanism, the seeding precipitation is specified as ice, but the presence of orography is not a prerequisite for the mechanism's occurrence. This natural cloud seeding is the focus of the current study, where hereafter the seeder-feeder mechanism and natural cloud seeding refer to ice particles falling from a cirrus cloud into
a lower-lying cloud or a lower-lying part of the same cloud, which is either liquid, ice or mixed-phase (Fig. 1). In a widened sense, the process of falling precipitation particles that feed on the hydrometeors in a lower part within the same cloud can also be understood as a seeder-feeder process (in-cloud seeder-feeder mechanism, Hobbs et al. (1980), see Fig. 1b).

Cirrus clouds, which act as seeder clouds in this study, can form either from freezing of liquid droplets or in-situ from homogeneous freezing of solution droplets or heterogeneous nucleation. Recent studies have suggested to classify cirrus clouds
accordingly, as liquid or in situ origin ice clouds (Luebke et al., 2013; Krämer et al., 2016; Luebke et al., 2016; Wernli et al., 2016; Gasparini et al., 2018; Wolf et al., 2019). The formation mechanism has been shown to influence clouds' microphysical properties (Luebke et al., 2016; Wolf et al., 2019).

Seeding ice crystals can have a large influence on cloud properties, because in the atmosphere, at temperatures warmer than $-38\,^\circ C$, ice can only be formed via heterogeneous nucleation on ice nucleating particles (Kanji et al., 2017). Once ice
particles are formed within the cloud or enter the cloud from outside, they grow by riming or vapour deposition (rapidly via the Wegener-Bergeron-Findeisen process, where ice crystals grow at the expense of liquid droplets, when the saturation ra-



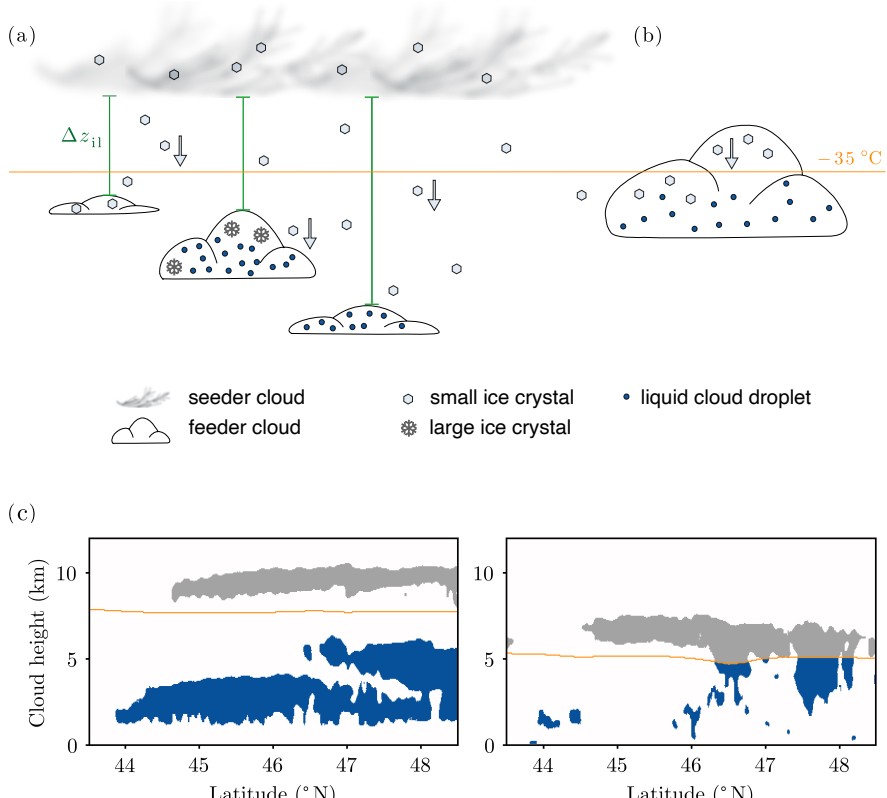

**Figure 1.** Sketch of the two seeder-feeder situations observed in this study. The orange lines depict the $-35\,^\circ$C isotherm, $\Delta z_{\mathrm{il}}$ is the distance between the lowest base of the cirrus cloud and the highest top of the cloud below. **(a)** Classical external seeder-feeder situation: a cirrus cloud ($T < -35\,^\circ$C) is detected at least $100\,$m above a cloud at $T > -35\,^\circ$C ($\Delta z_{\mathrm{il}} > 100\,$m). The latter cloud is termed *mixed-phase* cloud for simplicity, but could also be liquid or ice phase. **(b)** In-cloud seeder-feeder situation: the algorithm detects the cloud part above the $-35\,^\circ$C isotherm as a cirrus cloud, and the cloud part below as a mixed-phase cloud ($\Delta z_{\mathrm{il}} < 100\,$m). Ice crystal shapes are depicted according to Libbrecht (2005, Fig. 2). **(c)** Seeder-feeder situations as seen in the DARDAR data. Cirrus clouds above the $-35\,^\circ$C isotherm are depicted in grey, clouds below in blue. Left: exemplary plot of the classical external seeder-feeder situation (data from 29.05.2007), right: exemplary plot of only a mixed-phase or a cirrus cloud present (latitudes equatorwards of $46\,^\circ$N) and the in-cloud seeder-feeder situation (polewards of $46\,^\circ$N, data from 03.12.2010).



tio is between saturation with respect to water and ice (Wegener, 1911; Bergeron, 1935; Findeisen, 1938)), and can multiply through secondary ice production (Hallett-Mossop process (Hallett and Mossop, 1974; Mossop and Hallett, 1974; Mossop et al., 1974), frozen droplet shattering (Lauber et al., 2018), or ice-ice collisional breakup (Sullivan et al., 2018)). Thereby,
seeding ice crystals destabilize a cloud, which subsequently could glaciate and/or form precipitation. Because of the aforementioned enhancement processes in the ice phase, the seeder-feeder mechanism with seeding ice crystals is more efficient than the classical liquid seeder-feeder mechanism, and has been found to lead to a larger precipitation enhancement (Choularton and Perry, 1986).

For natural cloud seeding to take place, the ice crystals' survival during the sedimentation through a subsaturated layer of
air and into the lower cloud layer is crucial. Braham (1967) observed a spectacular case of ice crystals that survived a distance of 5 km in cloud-free air. This demonstrated the feasibility of natural cloud seeding (Hitschfeld, 1968; Locatelli et al., 1983). In a first theoretical study, Hall and Pruppacher (1976) found that "ice particles could survive distances of up to 2 km when the relative humidity with respect to ice was below 70 %". Natural cloud seeding through sedimenting ice crystals has been observed in a multitude of remote sensing and aircraft campaigns (Dennis, 1954; Hobbs et al., 1980, 1981; Locatelli et al.,
1983; Hobbs et al., 2001; Pinto et al., 2001; Fleishauer et al., 2002; Ansmann et al., 2008; Creamean et al., 2013) and has been studied in mostly idealized model simulations (Rutledge and Hobbs, 1983; Fernández-González et al., 2015; Chen et al., 2020), where it has been found to mainly enhance ice and precipitation formation.

Seifert et al. (2009) and Ansmann et al. (2009) estimated such an occurrence frequency of natural cloud seeding for their lidar field study datasets indirectly when aiming to exclude all seeded clouds. They simply defined all mixed-phase clouds that
had an ice cloud within 2 km above cloud top as a seeded ice cloud. For example, in Leipzig, about 10 % of ice-containing clouds at $-20\,°C$ were marked as seeded (ice containing clouds made up 90 % of the observations at that temperature). A more thorough, regional estimate of seeder-feeder occurrence frequency in the Arctic was derived by Vassel et al. (2019). Using radiosonde and radar data from Svalbard, they deduced the frequency of multilayer clouds as 29 %. Calculating the sublimation height of hexagonal plate ice crystals with a radius of 400 µm (radius meaning here: half of the maximum span
across the hexagonal face), 26 % of observations contained a seeding case.

Such field studies have begun to elucidate the frequency and thereby the importance of natural cloud seeding regionally, but a thorough estimate is still lacking. With global coverage and sensors increasingly capable of resolving clouds and their vertical distribution, satellite data offers an opportunity to fill the gap from single observations to whole-earth long-time observations to derive such a frequency estimate. Multilayer clouds can be investigated using CloudSat and CALIPSO data (e.g. Wang
et al. (2000); Mace et al. (2009); Das et al. (2017); Matus and L'Ecuyer (2017)). To provide an estimate of the natural cloud seeding frequency, sublimation calculations need to be combined with the seeder-feeder situation/multilayer cloud occurrence frequencies as done by Vassel et al. (2019).

In this study, we employ the DARDAR (ra*dar* li*dar*) satellite product that is based on CloudSat and CALIPSO data (Delanoë and Hogan, 2008, 2010b; Ceccaldi et al., 2013) and combine it with sublimation calculations to derive a frequency estimate
of seeder-feeder situations over Switzerland. Note that we consider as seeder clouds only cirrus clouds to ensure that the they contain ice. In the following Sect. 2, the DARDAR satellite product, our analysis and the sublimation calculations are





described. In Sect. 3.1, findings from the analysis of the DARDAR data are presented and discussed, followed by the results from the sublimation calculations in Sect. 3.2. Conclusions and an outlook are given in Sect. 4.

## 2   Methods and Data

### 2.1   Satellite data


The DARDAR satellite data product used in this study is based on radar, lidar and infrared radiometer data from the CloudSat and CALIPSO satellites. The satellites were launched jointly on 28 April 2006 into the A-Train or Afternoon Constellation, a coordinated group of satellites in a sun-synchronos polar orbit (Stephens et al., 2002). CloudSat has a cloud profiling radar on-board that senses cloud particles and detects precipitation (Stephens et al., 2008). CALIPSO carries the CALIOP lidar (Cloud-Aerosol Lidar with Orthogonal Polarization) and two passive sensors, a visible camera and a three-channel infrared radiometer (Winker et al., 2010). The two satellites are designed for their data to be combined: the lidar on CALIPSO is able to identify the thin upper layers of cirrus clouds that the radar on CloudSat misses (Winker et al., 2010), while the latter is able to look through thick clouds where the lidar beam is attenuated. Because of their joint operations and almost simultaneous time measurements, the two satellites provide novel ways to look at precipitation, aerosols and the vertical distribution of clouds (Gao et al., 2014; Hong and Liu, 2015; Naud et al., 2015; Stephens et al., 2018; Witkowski et al., 2018).



From the CloudSat and CALIPSO data, Delanoë and Hogan (2010b) developed the DARDAR (radar lidar) satellite product that provides cloud classification and ice cloud properties. It was developed further into a DARDAR v2 by Ceccaldi et al. (2013). DARDAR data is retrieved at $60\,\mathrm{m}$ vertical resolution up to an altitude of $25\,\mathrm{km}$ and a horizontal resolution of $1.4\,\mathrm{km}$ (Delanoë and Hogan, 2010a). Next to other cloud properties it contains a classification of the layer at each grid point with categories like clear sky, ice, liquid or supercooled clouds, aerosols, etc. as well as the retrieved effective ice crystal radius.


In this study, DARDAR-CLOUD v2.1.1 data (as described in Ceccaldi et al. (2013)) from April 2006 through October 2017 was used. Due to CloudSat's battery problems there is no data between April 2011 and April 2012 and merely Daylight-Only Operations mode data thereafter (Stephens et al., 2008; Witkowski et al., 2018; CloudSat radar status).

### 2.1.1   Analysis method


The study domain surrounds Switzerland (4°E to 12°E and 43.5°N to 48.5°N) and contains most of the Alps. Figure 2 shows the geographic distribution of all satellite tracks that go through the chosen domain. In order to evaluate the frequency of seeder-feeder situations four variables were created:



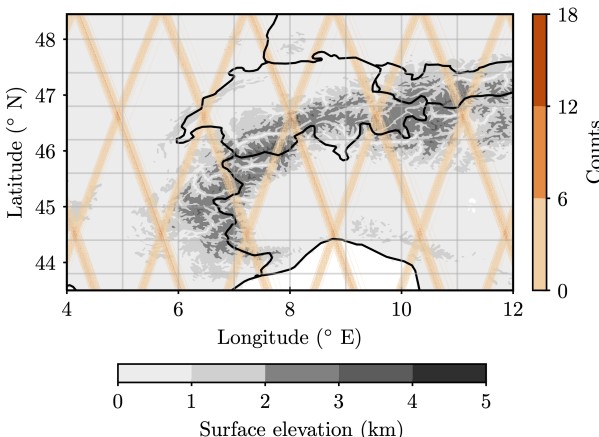

**Figure 2.** Geographical distribution of the satellite observations: number of tracks through each point within the study domain ($4°$E to $12°$E and $43.5°$N to $48.5°$N) over the whole time period analysed in this study (2006-2017).

| | |
|---|---|
| **frac_cov** (-) | The fraction of sky covered with a specific combination of cloud top and cloud base temperatures. |
| **icebase** (m) | The height (altitude above sea level) of the lowest cloud grid point with $T < -35\,°$C (lowest base of a cirrus cloud). |
| $\mathbf{\Delta z_{il}}$ (m) | The distance between the lowest cirrus cloud base and the highest top of the cloud below (in the following called mixed-phase cloud). |
| **reff** (µm) | The effective radius of ice crystals at the lowest cirrus cloud base. |

All variables were derived from a cloud mask, where the DARDAR categories 1, 2, 3 and 4 (ice, ice + supercooled, liquid
$> -35\,°$C and supercooled) were combined to simply signify the presence of cloud layers. This cloud mask was found to
be noisy and was therefore filtered (using a median filter over the surrounding $7 \times 7 \times 7$ points cube). For *icebase*, $\Delta z_{il}$,
and *reff* the cloud mask was combined with a temperature mask to differentiate between mixed-phase and cirrus clouds. In
this study, cirrus clouds are defined as clouds at temperatures lower than $-35\,°$C, and mixed-phase clouds are defined as all
clouds at temperatures warmer than $-35\,°$C. This is because liquid cloud droplets have been found to supercool to $-35\,°$C
before freezing homogeneously (Murray et al., 2010; Herbert et al., 2015). The temperature for homogeneous freezing of water
droplets is also often given as $-38\,°$C (Kanji et al., 2017). However, in tests preceding this study, a threshold of $-38\,°$C instead
of $-35\,°$C proved to have no evident impact on the results. Note that clouds termed mixed-phase could in principle be in the
liquid or ice phase in reality, depending on their history and the presence of ice nucleating particles (see Fig. 1a).

The combined cloud and temperature masks were applied to the altitude and effective ice crystal variable in the DARDAR
data to find the values at the lowest cirrus cloud base (for *icebase*, *reff* and $\Delta z_{il}$) and at the highest mixed-phase cloud top
(for $\Delta z_{il}$). Prior to this, the effective ice crystal radius was also filtered, for consistency. As a filter for the effective radius,
the vertical median with an extent of four pixels up and four pixels down from the one in question was applied, using only





**Table 1.** Variables used in the sublimation calculations as described in the text. For a comprehensive list, also see Table A1.

| Symbol | Long Name | Units |
|---|---|---|
| $C$ | capacitance of the ice particle | m |
| $D_\mathrm{v}$ | diffusivity of water vapour in air | $\mathrm{m^2\,s^{-1}}$ |
| $G$ | growth factor | $\mathrm{kg\,m^{-1}\,s^{-1}}$ |
| $m$ | mass of the ice particle | kg |
| $r$ | effective radius of the ice particle | m |
| $\rho_\mathrm{air}$ | air density | $\mathrm{kg\,m^{-3}}$ |
| $s$ | supersaturation with respect to ice | - |
| $v$ | fall speed of the ice particle | $\mathrm{m\,s^{-1}}$ |
| $z$ | height of the ice particle | m |

**Table 2.** Constants used in the sublimation calculations for a sphere as described in the text. For a comprehensive list, also see Table A2. Where the constants are different for a hexagonal plate or rosette crystal, they are given in Table A4 and A6.

| Symbol | Long Name | Value |
|---|---|---|
| $\alpha$ | coefficient for the velocity-mass-relation for cloud droplets (Seifert and Beheng, 2006, Table 1) | $3.75 \times 10^5 \ \mathrm{m\,s^{-1}\,kg^{-\beta}}$ |
| $\beta$ | coefficient for the velocity-mass-relation for cloud droplets (Seifert and Beheng, 2006, Table 1) | $2/3$ |
| $\gamma$ | coefficient for the velocity-mass-relation for cloud droplets (Seifert and Beheng, 2006, Table 1) | $1$ |
| $\rho_\mathrm{air,0}$ | reference density of air | $1.225 \ \mathrm{kg\,m^{-3}}$ |
| $\rho_\mathrm{i}$ | density of ice | $0.92 \times 10^3 \ \mathrm{kg\,m^{-3}}$ |

those pixels where the unfiltered cloud mask detected a cloud. For $\Delta z_\mathrm{il}$, the altitude of the highest mixed-phase cloud top was substracted from the altitude of the lowest cirrus cloud base. Finally, the dataset was saved on a grid with a resolution of

$0.005° \times 0.005°$ with no quality loss compared to the original DARDAR data. During regridding, areas containing no satellite tracks were set to missing data, to be able to derive the total number of observations later on.

## 2.2   Ice crystal sublimation calculations

Environmental parameters such as the air density, air temperature and the relative humidity determine the ice crystal sublimation rate and fall velocity. For these parameters, Hall and Pruppacher (1976) used the NACA standard profile, while Vassel

(2018) used mean values, and Vassel et al. (2019) used radiosonde profiles in their calculations. Since the environmental conditions are primary determinants of the sublimation height, we chose the most detailed information available. Relative humidity and temperature were therefore taken from ERA5 reanalysis data from the European Centre for Medium Range Weather Fore-





cast (ECMWF, Hersbach et al. (2020)). From the DARDAR data *icebase*, $\Delta z_{il}$, and *reff* were used. Prior to calculations, the ERA5 data was regridded: vertically to match the DARDAR $60\,\text{m}$ resolution; horizontally points closest to the DARDAR

points were chosen. As only hourly ERA5 data was available, data from the hour closest to the entry time of the satellites into the study domain was used. The sublimation height was calculated individually for every point in every available track file where there was at least one cirrus cloud above a mixed-phase cloud present. The algorithm is based on work in Vassel (2018). It was applied to three different shapes of ice crystals, namely spheres, hexagonal plates and bullett rosettes. These three were chosen to sample ice crystal properties, e.g. to span the possible range of terminal velocities. In particular, bullet rosettes have

been found to be one of the most abundant shapes in cirrus clouds (Lawson et al., 2019; Heymsfield and Iaquinta, 2000). And ice crystals have been found to evolve into spherical shape while sublimating (Nelson, 1998), which makes these ideal shapes to use. Additionally, the computations were run for plate like ice crystals, which experience intermediate drag and can also occur in cirrus clouds (Libbrecht, 2005), to include an ice crystal type used in Vassel et al. (2019). The equations shown refer to the spherical particle. Information for the computations using hexagonal plates and bullett rosettes is given in Tables A3 and

A4 in the Appendix.

The sublimation algorithm was applied in $0.01\,\text{s}$ timesteps ($\text{d}t$) as follows, where the initial height of the ice particle was *icebase*. The variables and constants used are given in Tables 1 and 2. The mass of the ice crystal was calculated from the radius:

$$m[0] = \frac{4}{3}r[0]^3\rho_i\pi \tag{1}$$

For a sphere, the capacitance of the ice particle is simply equal to the radius at timestep $i$ (Lohmann et al., 2016, pg. 240):

$$C = r[i] \tag{2}$$

Following Lamb and Verlinde (2011), the change in mass is

$$\text{d}m = 4\pi C[i]\rho_i G[i]s[i]f[i]\text{d}t \tag{3}$$

which was used to time step mass and radius of the ice crystal:

$$m[i+1] = m[i] + \text{d}m \tag{4}$$

$$r[i+1] = \sqrt[3]{\frac{3m[i+1]}{4\rho_i\pi}} \tag{5}$$

using the ventilation factor $f$ determined from Eq. (A5). The fall speed is calculated following Seifert and Beheng (2006), with coefficients given in Table 2, and used to timestep the height of the particle:

$$v[i+1] = \alpha m[i+1]^\beta\left(\frac{\rho_{\text{air},0}}{\rho_{\text{air}}}\right)^\gamma \tag{6}$$

$$z[i+1] = z[i] - v[i+1]\cdot\text{d}t \tag{7}$$



Equations used to generate the values needed in the above equations are given in Appendix A, with additional variables and constants in Tables A1 and A2.

In the calculations, radiative heat transfer to and from the ice particles was ignored since Hall and Pruppacher (1976) found that it "is only of secondary importance in determining [an ice particle's] survival distance in subsaturated air". While the
calculations are based on a scheme developed in Vassel (2018), here additional factors such as the ventilation factor and the temperature dependency in the dynamic viscosity were added. Furthermore, Vassel et al. (2019) used mass-diameter relations and fall speed derived in Mitchell (1996), which in this study are taken from Pruppacher and Klett (2010), Heymsfield and Iaquinta (2000) and Seifert and Beheng (2006) due to the differing ice crystal types used here.

The timestepping script was set to run for a day, but was stopped when the particle had reached Earth's surface or sublimated
(zero mass or a radius less than $10^{-8}$ m). The sublimation height was returned and compared to the height of the mixed-phase cloud top, which was derived from *icebase* and $\Delta z_{il}$ in the DARDAR data. When the sublimation height was lower than the height of the mixed-phase cloud top, the ice crystals at that grid point were marked as seeding.

These calculations present a conservative estimate. In reality, ice crystals have a size distribution. The large ice crystals within a distribution survive longer sedimentation distances than the ones with the effective radius, for which the survival is
calculated. Also, the effective radius of ice crystals is underestimated in DARDAR v2 compared to the newer version v3 (which is not available yet), by 5 % to as much as 40 % (Cazenave et al., 2019).

## 3 Results and Discussion

### 3.1 DARDAR data

#### 3.1.1 Distribution of distances between ice and mixed-phase cloud layer

Figure 3 shows the average frequency of $\Delta z_{il}$, the distance between the cirrus and mixed-phase cloud, within the DARDAR data set (as described in Sect. 2.1.1, any cloud at temperatures $> -35\,°C$ is termed mixed-phase in this study). It can be understood as the average distribution of $\Delta z_{il}$ within a unit area. 68 % of all measurements do not show a cirrus-mixed-phase cloud distance at all. In those cases, either only clouds of one category were present, or none at all (30 % of the measurements are cloud free). 32 % of the measurements contain both a cirrus and a mixed-phase cloud simultaneously. Tailoring this result
to the sedimentation of ice crystals from a cirrus cloud, 77 % of the measurements that detect a cirrus cloud also detect a lower mixed-phase cloud.

In 56 % of these cases (18 % in total), $\Delta z_{il}$ is smaller than 100 m. This may either be the case when the cirrus and the mixed-phase cloud are truly separated by a small distance, or when the two differently classified layers are actually part of the same cloud. From the construction of the classification algorithm, the latter would be the case when the $-35\,°C$ isotherm
intersects the cloud. This case is illustrated in Fig. 1b. In contrast to Mace et al. (2009) and Vassel et al. (2019), our algorithm does not require a cloud-free layer in between the mixed-phase and the cirrus cloud, so we also observe a potential for in-cloud seeding. However, clouds connected by sedimenting ice would also be seen as a cirrus cloud with a very small or no distance to





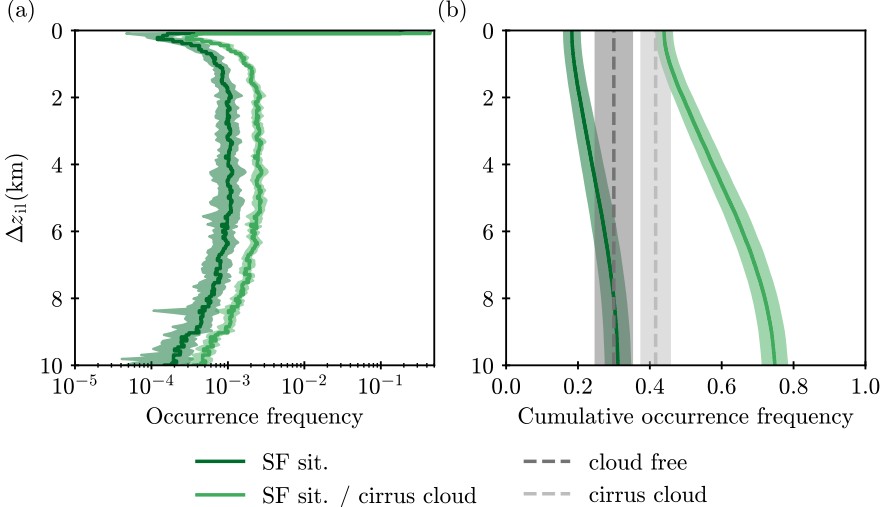

**Figure 3. (a)** Occurrence frequency of seeder-feeder situations (SF sit.) with respective $\Delta z_{il}$ as a fraction of measurements (dark green) or cirrus cloud measurements (light green). **(b)** Cumulative occurrence frequency. For $\Delta z_{il}$ a vertical resolution of $60\,\mathrm{m}$ is used. For comparison, the fraction of measurements with at least one cirrus cloud (light grey) and with a cloud free atmposphere (dark grey) are given. Here and in the following data from all tracks in the study time (2006 to 2017) and within the study domain were used (2210 satellite tracks). The total number of measurements is 1440312, with 853833 measuring $\Delta z_{il}$ and 355331 measuring cirrus clouds. The shaded areas visualize the standard deviation of interanual variablility. Note that $\Delta z_{il} = 0\,\mathrm{m}$ is at the base of the lowest cirrus cloud layer with $T < -35\,^{\circ}\mathrm{C}$.

the next mixed-phase cloud in our analysis. Ansmann et al. (2009) observed ice virga between the seeder and the feeder cloud and Mace et al. (2009) also mentioned this as a cause of misclassification in their study. Of course, in cases where the $-35\,^{\circ}\mathrm{C}$
isotherm lies within the cirrus cloud, there could be another mixed-phase cloud underneath. The distance to this second cloud does not appear in our analysis.

The other half of the cases ($\Delta z_{il} > 100\,\mathrm{m}$) represents the classical external seeder-feeder situation, with a cirrus cloud clearly separated from a mixed-phase cloud below (see Fig. 1a). The $\Delta z_{il}$ are distributed equally between $2000\,\mathrm{m}$ and $6000\,\mathrm{m}$, increase for smaller and decrease for larger $\Delta z_{il}$. The smaller frequencies at $\Delta z_{il} < 2000\,\mathrm{m}$ are due to the few possibilities for
both cirrus and mixed-phase cloud to be located close to the $-35\,^{\circ}\mathrm{C}$ isotherm. Because the cirrus cloud frequency decreases for large heights, the $\Delta z_{il}$ occurrence frequency decreases as well for $\Delta z_{il} > 6000\,\mathrm{m}$. Generally speaking, $\Delta z_{il}$ increases with increasing upper cloud height (see Fig. B2).

In this distribution and in the following analysis, the effect of vertical wind shear of the horizontal wind cannot be taken into account, because the satellite retrieval only obtains instantaneous profiles, without any information on their temporal
development. In the time that ice crystals need to sediment distances of a few kilometers, undoubtedly the clouds in question move relative to each other when wind shear is present. However, this movement can go in both ways, either removing or





creating a multilayer cloud situation. On average, these two effects are expected to cancel out, so that the results with and without considering wind shear should be similar.

Our results for multi-layer cloud occurrence frequency are similar to but smaller than the ones given in the literature. In
their analysis of CALIPSO and CloudSat data, Mace et al. (2009) estimated the global occurrence of multiple layers to be 24 %. Wang et al. (2000) derived an estimate of 42 % from a radiosonde dataset. Of course the domain around Switzerland in this study is not expected to reproduce the global average, but Fig. 17a in Mace et al. (2009) and Fig. 5 in Wang et al. (2000) show average frequencies for Switzerland that are similar to the global average frequency. Using CloudSat and CALIPSO data as well, Matus and L'Ecuyer (2017) found an average multilayer cloud fraction of about 25 % for the mid latitudes of
Switzerland. The layers derived from radiosonde data by Wang et al. (2000) are much thinner than the ones found with remote sensing, possibly because large sedimenting particles cause multiple thin layers to be identified as one large layer by the radar (Mace et al., 2009). One might therefore expect that the results from this current satellite study are closer to the ones from Mace et al. (2009) and Matus and L'Ecuyer (2017). Most importantly, the present study only looks at multiple layer occurrence between cirrus and mixed-phase clouds, which is lower than the total multilayer occurrence frequency. As a proxy for this, one
might use the relative occurrence frequency of low with high and mid with high clouds from Mace et al. (2009) (about 70 % and 10 %), relative to their overall multilayer occurrence frequency of 24 %. Their resulting absolute high with low or mid cloud layer occurrence frequency is then approximately 20 %. The result for two cloud cases in this study of 15 % is smaller than the value derived by Mace et al. (2009), although they used even more restrictive conditions for their classification of multiple layers, requiring almost 1 km of cloud free space in between them. As mentioned before, the in-cloud seeder-feeder
situations provide no information on the occurrence of mixed-phase layers below, hiding possible two cloud cases.

### 3.1.2 Effect of topography

A geographical difference in cloud cover could be expected from the differing impacts that weather regimes have on different European regions in general (Pasquier et al., 2019; Grams et al., 2017). The study domain contains locations with a large range of surface altitudes (see Fig. 2). One could imagine the $\Delta z_{il}$ to be smaller in the Alps than over the Swiss Plateau, simply
because of a thinner troposphere over orography. Also the orographic forcing would be expected to increase cloud cover. For an analysis of topographical influence, we split the dataset by surface altitudes above or below 1 km and analyse the distribution of $\Delta z_{il}$, shown in Table 3. The difference in the fraction of distances larger than 100 m between locations with a topography higher or lower than 1 km is less than 1 %. The distribution of total $\Delta z_{il}$ between mountaineous terrain and flat land reproduces the distribution of measurements (about 30 % are taken over orography higher than 1000 m and about 70 % over terrain lower
than 1000 m, not shown). Contrary to what we expected, we find no topographical effect in the distribution of $\Delta z_{il}$ (see also Fig. B1).

### 3.1.3 Effect of season and time of day

Table 3 also contains the results of a climatological analysis of $\Delta z_{il}$. Winter measurements have more multi-layer clouds according to our definition than summer measurements. The relative increase is similar for the smaller ($\Delta z_{il} < 100$ m) and the



**Table 3.** $\Delta z_{il}$ climatology: Fraction (%) of $\Delta z_{il}$ smaller than $100\,\mathrm{m}$ and larger than $100\,\mathrm{m}$ in all measurements with the specified surface height, for summer vs. winter and day vs. night (Julian days $\geq 106$ and $< 289$ are summer, hours $\geq 6$ and $< 18$ are day). $\Delta z_{il}$ up to $12\,\mathrm{km}$ in length were evaluated.

| Season | Time of day | Whole domain | | Surface $< 1\,\mathrm{km}$ | | Surface $> 1\,\mathrm{km}$ | |
|---|---|---|---|---|---|---|---|
| | | $\Delta z_{il} < 100\,\mathrm{m}$ | $\Delta z_{il} > 100\,\mathrm{m}$ | $\Delta z_{il} < 100\,\mathrm{m}$ | $\Delta z_{il} > 100\,\mathrm{m}$ | $\Delta z_{il} < 100\,\mathrm{m}$ | $\Delta z_{il} > 100\,\mathrm{m}$ |
| All | All | 18 | 13 | 18 | 13 | 19 | 13 |
| | Day | 18 | 13 | 18 | 13 | 19 | 13 |
| | Night | 18 | 14 | 18 | 14 | 18 | 14 |
| Summer | All | 17 | 11 | 16 | 12 | 17 | 12 |
| | Day | 17 | 11 | 16 | 12 | 17 | 12 |
| | Night | 16 | 12 | 16 | 12 | 17 | 12 |
| Winter | All | 21 | 15 | 21 | 14 | 21 | 15 |
| | Day | 21 | 13 | 20 | 14 | 21 | 14 |
| | Night | 21 | 17 | 21 | 17 | 21 | 17 |

larger distances ($\Delta z_{il} > 100\,\mathrm{m}$). In particular, winter nights have the highest fraction of multiple layer cloud measurements. Multiple cloud layers are about $23\,\%$ more frequent in winter nights than in summer nights, mostly due to an increase in $\Delta z_{il}$ larger than $100\,\mathrm{m}$. There is no noticeable difference in frequencies during day and night.

The simplest explanation for the increased frequency of multi-layer clouds in winter measurements is simply an increased cloud cover in winter. To see whether this is a robust finding, it was tested with the CALIPSO-GOCCP dataset (Chepfer et al., 2010, 2013). With a comparison of *frac_cov* from DARDAR vs. CALIPSO cloud cover data (Fig. C1), the two datasets were found to mostly agree. Therefore, the CALIPSO dataset can be used to validate the hypothesis of an increased cloud cover in winter. Indeed in CALIPSO, total winter cloud cover is higher over almost the whole domain (Fig. C1c). The increase of cloud cover in winter is strongest for low and high clouds (low clouds: pressure $> 680\,\mathrm{hPa}$, height $< 3.2\,\mathrm{km}$, high clouds: pressure $< 400\,\mathrm{hPa}$, height $> 6.5\,\mathrm{km}$ in the CALIPSO data; not shown). This confirms the finding that in winter we see an increase in both small and large $\Delta z_{il}$. In addition, *icebase* is lower in winter (not shown), in particular for $\Delta z_{il} < 100\,\mathrm{m}$, which also increases the number of $\Delta z_{il}$.

### 3.1.4 Ice crystal effective radius and cirrus cloud origin

The DARDAR dataset provides the mean effective ice crystal radius, which we use in our sublimation calculations. In Fig. 4, the size distribution is displayed by the ice crystals' occurrence height, namely the lowest cirrus cloud base heights (*icebase*). The ice crystal size range, between $25\,\mu\mathrm{m}$ and $60\,\mu\mathrm{m}$ in radius, agrees with the one found in another DARDAR study by Hong and Liu (2015). It is also within the range from $1\,\mu\mathrm{m}$ to $100\,\mu\mathrm{m}$ that Krämer et al. (2009) find for cirrus clouds in aircraft campaigns.





There is a visible trend for smaller ice crystals at higher altitudes. This again agrees with Hong and Liu (2015) and Heymsfield et al. (2013), who find that ice crystal size decreases with decreasing temperature. An interesting feature in Fig. 4a is that while the shape of the distribution is rather symmetrical around this trend, large ice crystals abruptly stop appearing at heights larger than about $9.5\,\mathrm{km}$. This hints to two modes within the size distribution. These have been found in earlier studies, and have lately been linked to the different origins of cirrus clouds by Luebke et al. (2013), Luebke et al. (2016), Krämer et al. (2016), Wernli et al. (2016), Gasparini et al. (2018), and Wolf et al. (2019). These studies distinguish in situ origin cirrus clouds, which form by homogeneous nucleation of solution droplets or heterogenous nucleation of ice nucleating particles within the cirrus temperature range, and liquid origin clouds, which form from supercooled water droplets which are uplifted to the cirrus temperature range and freeze either heterogeneously at warmer temperatures or predominantly homogeneously at temperatures below $-35\,^\circ\mathrm{C}$. The two types mostly differ in their ice water content and the ice crystal size, with both being larger for liquid origin cirrus clouds (Luebke et al., 2016).

We split the dataset into one part with $\Delta z_{\mathrm{il}} > 100\,\mathrm{m}$ and one with $\Delta z_{\mathrm{il}} < 100\,\mathrm{m}$ as a proxy for the two cloud origins: in situ origin cirrus have large distances to the next underlying mixed-phase cloud, while liquid origin cirrus appear close to the $-35\,^\circ\mathrm{C}$ isotherm. This separation indeed produces two different modes, as can be seen in Fig. 4b and 4c. Figure 4b displays larger ice crystals, from $\approx 35\,\mu\mathrm{m}$ to $\approx 90\,\mu\mathrm{m}$ at cirrus cloud base heights from $4500\,\mathrm{m}$ to $9500\,\mathrm{m}$, with an abrupt decrease in occurrence frequency at cirrus cloud base heights higher than $9500\,\mathrm{m}$. The decrease at the maximum cirrus cloud base height is associated with $\Delta z_{\mathrm{il}} < 100\,\mathrm{m}$ (see Fig. B2). On the other hand, Fig. 4c displays smaller crystals, from $\approx 30\,\mu\mathrm{m}$ to $\approx 60\,\mu\mathrm{m}$, over a larger cirrus cloud height range, from roughly $6\,\mathrm{km}$ to $13\,\mathrm{km}$. Here the trend of smaller ice crystals at larger cirrus cloud heights is obvious. Figure 4 confirms the distinction between in situ and liquid origin cirrus clouds as proposed e.g. by Krämer et al. (2016). It also confirms the finding from Luebke et al. (2016) that liquid origin cirrus clouds are composed of ice crystals.

There are a few caveats to this result. First, by the construction of the classification algorithm, in situ cirrus clouds are sampled for the ice crystal radius at their base, while liquid origin clouds are sampled in the interior. However, this difference is expected to have the opposite effect of what we observed. At the cloud bases, the ice crystals are expected to be larger than in their middle (Miloshevich and Heymsfield, 1997; Heymsfield and Iaquinta, 2000), simply because of larger particles sedimenting further down within a cloud. Secondly, the classification scheme only has liquid origin clouds in the $\Delta z_{\mathrm{il}} < 100\,\mathrm{m}$ part, while liquid origin clouds that have been uplifted entirely to heights above the $-35\,^\circ\mathrm{C}$ isotherm are present in the second, in situ origin cirrus part of the dataset ($\Delta z_{\mathrm{il}} > 100\,\mathrm{m}$), if such a lifting occurs. This erroneous classification has already been noted by Gasparini et al. (2018). However, Fig. 4c displays only one mode, missing any signal of the mode present in the $\Delta z_{\mathrm{il}} < 100\,\mathrm{m}$ part of the dataset (see Fig. 4b). This suggests that the influence of the liquid origin on the microphysical properties of the cirrus clouds is lost once the clouds are lifted, for example because the large ice crystals sediment out, or that lifting of entire clouds above the $-35\,^\circ\mathrm{C}$ isotherm hardly ever occurs. Wernli et al. (2016), who investigated the frequency of the formation pathways in a trajectory-based analysis, already noted that ice crystal sedimentation and cloud turbulence could "potentially alter the local cirrus characteristics and 'confuse' the simple categorization". This seems to be the case with the data presented here, or otherwise the data suggests that liquid origin clouds are hardly ever lifted entirely above the $-35\,^\circ\mathrm{C}$ isotherm.

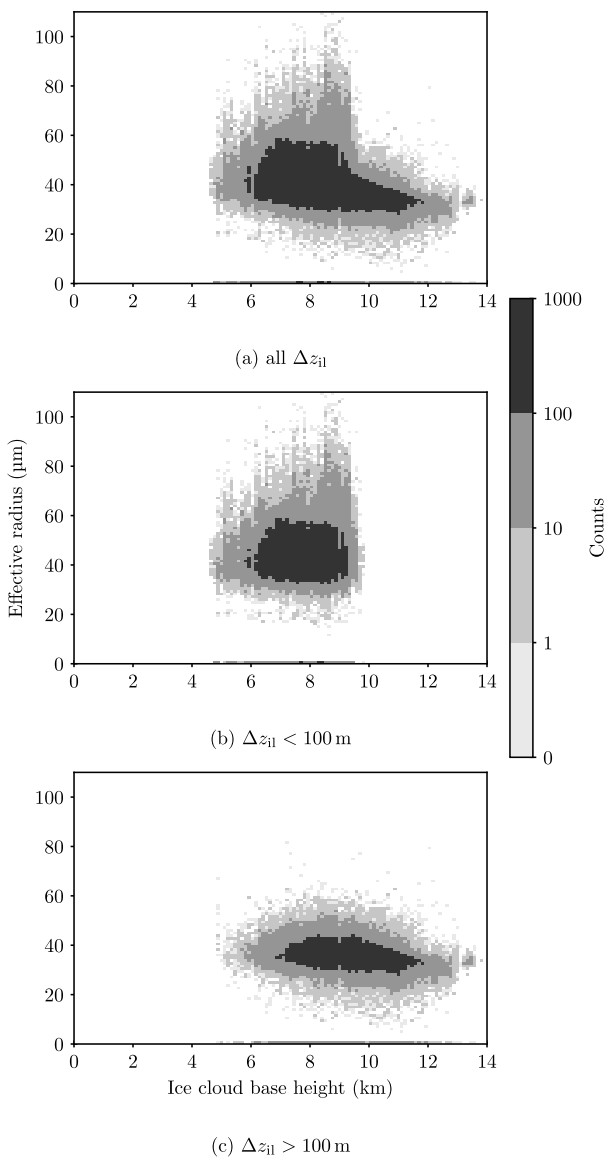

**Figure 4.** Distribution of *reff*. **(a)** For all multilayer clouds. **(b)** Only those data points with a distance $< 100\,\mathrm{m}$ to the next mixed-phase cloud top. **(c)** Only those data points with a distance $> 100\,\mathrm{m}$ to the next mixed-phase cloud top.

In a broader context, the results in Fig. 4 show that satellite data, in particular the DARDAR dataset, are valid means to explore the classification of cirrus clouds into liquid and in situ origin further, as it has been called for by Wolf et al. (2019).

305    Note that the ice crystals radii, the cirrus cloud base heights and the $\Delta z_{il}$'s span a wide range of values (see Fig. 3 and 4). Therefore, sublimation calculations needed to be applied to each instance individually, as detailed in the next section.



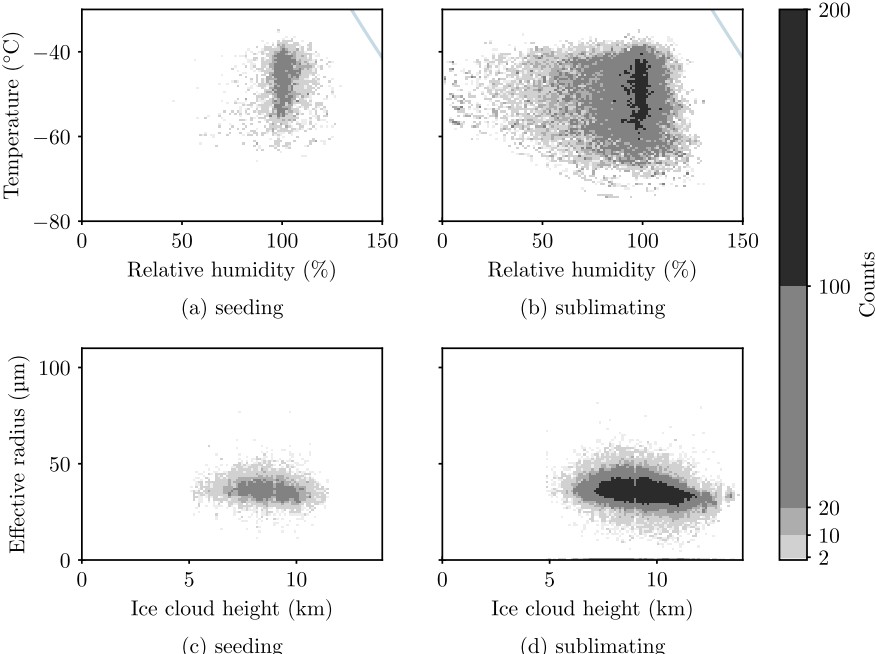

**Figure 5.** Environmental conditions at cirrus cloud base. Absolute frequency of temperature as a function of relative humidity with respect to ice at cirrus cloud bases with $\Delta z_{il} > 100\,\text{m}$ and **(a)** where spherical ice crystals survive the sedimentation and seed the lower cloud, **(b)** where spherical ice crystals sublimate before reaching the mixed-phase cloud. The light blue line depicts saturation with respect to water. Absolute frequency of effective ice crystal radius at cirrus cloud base as a function of cirrus cloud height with $\Delta z_{il} > 100\,\text{m}$ and **(c)** where spherical ice crystals survive the sedimentation and seed the lower cloud, **(d)** where spherical ice crystals sublimate before reaching the mixed-phase cloud. The sum of **(c)** and **(d)** is displayed in Fig. 4c. For improved readability the colorbar label for bin 1 is not shown.

## 3.2 Sublimation between cloud layers

As described in Sect. 2.2, the sublimation calculation was applied to each grid point within the DARDAR data that had a cirrus cloud present above a mixed-phase cloud layer, using DARDAR and ERA5 data as input. The sublimation height of the ice 310 crystals was calculated three times, assuming spherical ice crystals, plates and bullet rosettes. If the sublimation height was lower than the mixed-phase cloud top, the case was marked as a seeder-feeder situation.

### 3.2.1 Variation of survival with environmental parameters

For the evaluation of the survival chance, only cases with $\Delta z_{il} > 100\,\text{m}$ were taken into account. Distances smaller than $100\,\text{m}$ represent the in-cloud seeder-feeder mechanism, where ice crystals fall through saturated or supersaturated cloudy air only 315 before interacting with other hydrometeors. Comparing Fig. 5a and 5b, one can see the effect of temperature and relative humidity: ice crystals only reach the lower cloud if $\text{RH}_i > 90\,\%$. Only those starting at temperatures warmer than $-65\,°\text{C}$





seed. At lower temperatures, the ice crystals sublimate, even if the air was supersaturated at the start of the sedimentation. Note that due to data storage constraints, we can only show the impact of the temperature and relative humidity at the starting cirrus cloud base height on the falling ice crystals. But height resolved ERA5 data of temperature and relative humidity was used for the calculations. These starting values can be seen as proxies for the values during sedimentation, but for large sedimentation distances of up to about $5\,\mathrm{km}$, the starting values are not representative. Vassel et al. (2019) conducted a sensitivity study with relative humidities varying by $\pm 5\,\%$, but this variation is rather small. In this, their resulting seeding fraction does not change substantially. However, the relative humidity variations over the distances traveled by ice crystals in our calculations can exceed $5\,\%$ substantially.

Figure 5c shows that ice crystals do not survive the fall from cirrus cloud base heights above $11\,\mathrm{km}$. We attribute this to the fact that high cirrus cloud bases correspond to large distances to lower lying mixed-phase clouds that ice crystals are less likely to survive. Both Hall and Pruppacher (1976) and Vassel et al. (2019) identified the ice crystal size as important determinant for ice crystal survival. Here, we find that ice crystals with radii smaller than $30\,\mu\mathrm{m}$ usually do not survive the sedimentation. On the other hand also larger ice crystal sizes, above $50\,\mu\mathrm{m}$, do not guarantee a successful seeding. Note that we only evaluate the mean ice crystal size is used in this study so that the large spread which occurs in ice crystal size distributions is not represented.

For both the analysis of environmental parameters and DARDAR variables on ice crystal survival, the results assuming ice crystals to be plates and bullet rosettes are similar to those presented in here. One marked difference is that crystals starting in a subsaturated environment with respect to ice sublimate and do not seed when assuming them to be plates or bullet rosettes (see Fig. B3).

A comparison to literature data is difficult because the assumptions vary greatly between studies. Hall and Pruppacher (1976) compute sublimation heights for ice particles with an initial radius of $160\,\mu\mathrm{m}$, at fixed relative humidities with respect to ice between $30\,\%$ and $90\,\%$. Their spherical ice particles sublimated at distances of $1\,\mathrm{km}$ to $4\,\mathrm{km}$ from the starting altitude of about $9\,\mathrm{km}$. The relative humidities that we find at the starting altitudes are similar to their range, as are our survival distances. Vassel et al. (2019) did not provide information on the distances between the cloud layers they studied. Preliminary work in Vassel (2018) contained the result of two exemplary sublimation calculations assuming constant temperature and relative humidity in the subsaturated layer. Her result is in line with the results presented in Fig. 6, where about $42\,\%$, $47\,\%$ or $64\,\%$ of cases with $\Delta z_{\mathrm{il}} = 500\,\mathrm{m}$ lead to successful seeding (for rosettes, plates and spheres respectively).

### 3.2.2 Influence of the ice crystal shape

The fraction of $\Delta z_{\mathrm{il}}$ with successful seeding is shown in Fig. 6 for plates, spherical ice crystals and bullet rosettes. For $\Delta z_{\mathrm{il}} > 5\,\mathrm{km}$, there is a only a slight chance for ice crystals to survive the fall between the cirrus and the underlying mixed-phase cloud. For $\Delta z_{\mathrm{il}} = 2\,\mathrm{km}$ the survival rate of spherical ice crystals increases to $20\,\%$. Survival chances increase linearly, until $81\,\%$ of the spherical ice crystals cause seeding at a falling distance of $200\,\mathrm{m}$. Plate like ice crystals experience a larger drag force and therefore fall slower than spheres. As they have more time to sublimate during their slower fall, they are less likely to survive at any of the distances. This was also found by Hall and Pruppacher (1976), and is even more pronounced for bullet rosettes. Combining this with the respective $\Delta z_{\mathrm{il}}$ frequencies, Fig. 6 also displays the fraction of successful seedings in our





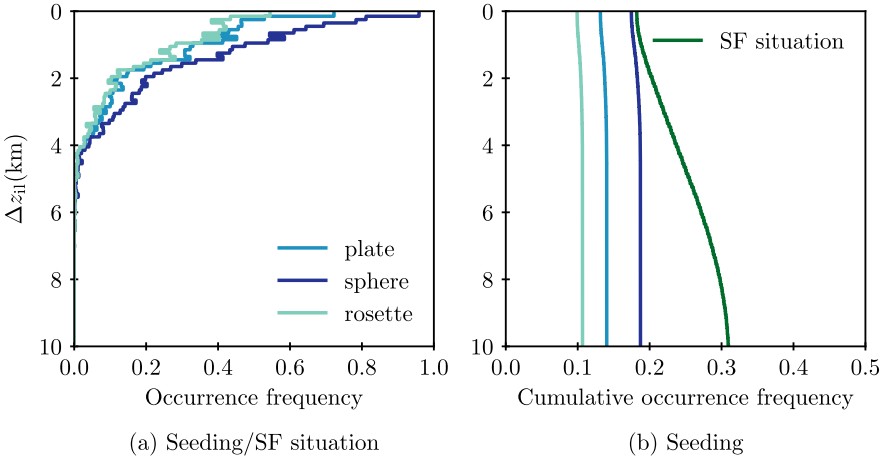

(a) Seeding/SF situation                    (b) Seeding

**Figure 6. (a)** Seeding cases per seeder-feeder situation. **(b)** Cumulative occurrence frequency of possible seeder-feeder situations (SF situation, green), and successful seeding assuming plate like, spherical and bullet rosette ice crystals. Note that $\Delta z_{il} = 0\,\mathrm{m}$ is at the base of the lowest cirrus cloud layer with $T < -35\,^{\circ}\mathrm{C}$.

measurements. In $14\,\%$ of the measurements, we see a seeder-feeder situation where plate-like ice crystals do not sublimate but can seed the lower lying cloud after sedimentation ($11\,\%$ for rosettes, and $19\,\%$ for spheres).

A surprising result for all ice crystal shapes is that the survival fraction for $\Delta z_{il} < 100\,\mathrm{m}$ is smaller than 1. As explained before, there is no subsaturated layer in this continuous cloud, so the sedimenting ice crystals should not sublimate at all. The
reason for the discrepancy most likely lies in the usage of two independent datasets in the classification of cloud layers and the calculation of ice crystal survival: the distance between the two layers and the cloud heights are taken from the DARDAR dataset, while the relative humidity was taken from ERA5. For example, the temperature profile in ERA5 over Switzerland is about $5\,^{\circ}\mathrm{C}$ colder than the one in the DARDAR data, which also originates from ECMWF. A reason for this discrepancy could not be found and it is not thought to change our findings significantly, but the discrepancy between the data sets should
be investigated further. One might correct for this by simply setting the survival fraction to 1 within the $\Delta z_{il} < 100\,\mathrm{m}$ bin, i.e. within cloud. However, we chose to leave the inconsistency as an estimate of the uncertainty associated with the seeding fractions given for larger distances.

In general, as stated before, the ice crystal radius and hence the survival fraction shown in Fig. 6 are conservative estimates. In particular, with the new DARDAR dataset (v3) (Cazenave et al., 2019), survival fractions are expected to be higher than
shown here for DARDAR v2, since the effective ice crystal radii are larger in the former (see Sect. 2). In their sublimation calculations, Vassel et al. (2019) use larger ice crystal radii of $100\,\mu\mathrm{m}$ for cirrus clouds as well. Additionally, there is the possibility of seeding by pre-activated particles even after the macroscopic ice crystal has sublimated, as described in Marcolli (2017). Some ice in pores or shielded pockets of these particles could survive the subsaturated air in between cloud layers and initiate new ice crystal formation once the particle reaches the supersaturated air in the lower cloud layer.





With the results presented here, one can comment on the method used in Seifert et al. (2009) to filter out ice clouds that were seeded. They simply reclassified any cloud with an ice cloud less than $2\,\mathrm{km}$ above as a liquid cloud. Given that Fig. 6 shows that only 10 to $20\,\%$ of ice crystals survive $\Delta z_{\mathrm{il}} = 2\,\mathrm{km}$, it is likely that Seifert et al. (2009) find too many seeded clouds. Finally, comparing to observations, the case of a survival of $\Delta z_{\mathrm{il}} = 5\,\mathrm{km}$, as the one case evaluated in Braham (1967), is rather unlikely according to our data.

## 4    Summary and conclusions

This study uses satellite data and sublimation calculations to establish the occurrence frequency of seeder-feeder cases over Switzerland. The seeder-feeder mechanism here refers to ice crystals that fall from a cirrus cloud into a lower cloud, where they act as seeds for the glaciation of clouds.

In the DARDAR data, we distinguish two situations: in $13\,\%$ of all (including clear-sky) measurement cases, distances
between the two cloud types are distributed uniformly between $100\,\mathrm{m}$ and $10\,\mathrm{km}$. This is the classical external seeder-feeder situation, where the seeding ice crystals fall through clear air between two clouds. In-cloud seeder-feeder situations are found to occur in $18\,\%$ of all measurements. In total, seeder-feeder cloud situations were found to occur in $31\,\%$ of all measurements. As the estimate only includes cases with a cirrus cloud as the seeder cloud, it underestimates the total seeder-feeder cloud situation occurrence frequency. The frequency was found to not vary with the differing topography in Switzerland. Seasonally,
winter nights exhibit the highest frequency of possible seeder-feeder situations due to an increased high cloud cover in winter and at night.

We find two modes for the ice crystals size at the base of cirrus clouds. These correspond to in situ and liquid origin cirrus clouds, which confirms the new classification scheme for cirrus clouds (Luebke et al., 2013, 2016; Krämer et al., 2016; Wernli et al., 2016; Gasparini et al., 2018; Wolf et al., 2019).

In sublimations calculations we found that a significant number of ice crystals reached the lower cloud layers. $20\,\%$ of ice crystals survived distances of $2\,\mathrm{km}$ when assuming that they were spherically shaped. Assuming plate-like crystals or bullet rosettes in the calculations, only about $10\,\%$ of them survived $2\,\mathrm{km}$ distances. On the one hand, this clearly shows that natural cloud seeding occurs regularly over Switzerland. On the other hand, it demonstrates that in these calculations, the distinction between ice crystal shapes is critical, in contrast to the small ice crystal shape impact found in Vassel et al. (2019).

We found that ice crystals only survive the fall between cloud layers when the relative humidity with respect to ice at cirrus cloud base is larger than $90\,\%$, while temperature seems to be of secondary importance. In terms of the ice crystal radius, ice crystals with effective radii smaller than $30\,\mu\mathrm{m}$ mostly sublimate before reaching the lower cloud layer. On the other hand, larger ice crystal sizes, above $50\,\mu\mathrm{m}$, do not guarantee a survival.

Taking a broader perspective, this study demonstrates that satellite data is a viable mean to explore cloud distributions also
in regional settings. It can be combined with timestepping calculations to study processes on which the satellite data, which is merely a snapshot in time, provides no information by itself.





Of course the scope of this work could be broadened in the future. This study focuses on natural cloud seeding that originates from cirrus clouds, but seeding ice crystals can also sediment from mixed-phase clouds. Additionally, multilayer clouds interact in other ways, for example via radiation (Christensen et al., 2013; Vassel, 2018). Moreover, seeing that natural cloud seeding
occurs over Switzerland, the global distribution of seeder-feeder cloud situations and the seeding frequency are an interesting next goal of study. Differences in the global distribution of multilayer clouds have already been demonstrated (Mace et al., 2009), and Ansmann et al. (2009) observed an increase in in-cloud seeding frequency in their data from the tropics compared to data from the mid latitudes (Seifert et al., 2009), so a thorough study of global natural cloud seeding frequency promises to be interesting. The satellite data analysis within this study can easily be extended to a global dataset. Solely the sublimation
calculations could not be applied to each measurement point in such a large dataset, but instead the seeding situations could be classified and sublimation calculations could be applied to the classes in a representative fashion. Future work could sample the whole range in ice crystal size distributions instead of only using the mean size to represent the distribution as done in this study.

We show that natural cloud seeding is a widespread phenomena over Switzerland. This hints to a large potential for nat-
ural cloud seeding to alter cloud properties and thereby influence Earth's radiative budget and water cycle, which should be investigated. We do so in a companion paper, using sensitivity simulations with the regional climate model COSMO.

*Code and data availability.* Analysis and plotting scripts are archived at https://doi.org/10.5281/zenodo.3987754. Generated data is archived at https://doi.org/10.5281/zenodo.3987757. DARDAR-CLOUD data can be obtained from the AERIS/ICARE Data and Services Center, ftp: //ftp.icare.univ-lille1.fr/SPACEBORNE/MULTI_SENSOR/DARDAR_CLOUD/ (last access: 5 October 2020). Copernicus Climate Change
Service (C3S) (2017): ERA5: Fifth generation of ECMWF atmospheric reanalyses of the global climate. Copernicus Climate Change Service Climate Data Store (CDS), 7 November 2019.





**Table A1.** Variables used in the sublimation height calculation (addition to Table 1).

| Symbol | Long Name | Units |
|--------|-----------|-------|
| $e$ | saturation of vapour pressure in air | Pa |
| $e_{\mathrm{sat,i}}$ | saturation vapour pressure with respect to ice | Pa |
| $e_{\mathrm{sat,w}}$ | saturation vapour pressure with respect to water | Pa |
| $L_{\mathrm{s}}$ | latent heat of sublimation | $\mathrm{J\,mol^{-1}}$ |
| $\mu$ | dynamic viscosity | $\mathrm{kg\,m^{-1}\,s^{-1}}$ |
| $N_{\mathrm{Re}}$ | Reynolds number | - |
| $p$ | pressure | Pa |
| RH | relative humidity | % |
| $\rho_{\mathrm{air}}$ | air density | $\mathrm{kg\,m^{-3}}$ |
| $T$ | temperature in K | K |
| $T_{\mathrm{°C}}$ | temperature in °C | °C |

## Appendix A: Sublimation calculations

Here we detail the equations used in the sublimation calculations. Additional variables and constants used are given in Tables A1 and A2. Where they differ, equations and constants used for the computations for hexagonal plates are given in Tables A3 425 and A4.

At each timestep $i+1$ the barometric formula was applied to find the pressure corresponding to the height of the ice particle:

$$p = p_0 \left( \frac{T_{\mathrm{b}}}{T_{\mathrm{b}} + L_{\mathrm{b}} \cdot z[i]} \right)^{\frac{g M_{\mathrm{air}}}{R L_{\mathrm{b}}}} \tag{A1}$$

The density of the air surrounding the particle was calculated using the ideal gas law. The saturation vapour pressure of water with respect to ice and water was derived with the Magnus formula. And together with the relative humidity from the ERA5 430 data (given with respect to water), the supersaturation with respect to ice was calculated. The diffusivity of water vapour in air was calculated following Hall and Pruppacher (1976, Eq. 13):

$$D_{\mathrm{v}} = 0.211 \times 10^{-4} \left( \frac{T}{T_0} \right)^{1.94} \frac{p_0}{p} \tag{A2}$$

From this, the growth factor was determined following Lamb and Verlinde (2011, pg. 328):

$$G = \frac{1}{\frac{\rho_{\mathrm{i}} R T}{M_{\mathrm{w}} D_{\mathrm{v}} e_{\mathrm{sat,i}}} + \frac{\rho_{\mathrm{i}} L_{\mathrm{s}}}{M_{\mathrm{w}} k_{\mathrm{T}} T} \cdot \left( \frac{L_{\mathrm{s}}}{R T} - 1 \right)} \tag{A3}$$

which uses the latent heat of sublimation (valid between 236 K and 273.16 K, Lohmann et al. (2016)):

$$L_{\mathrm{s}} = 46782.5 + 35.8925 \cdot T - 0.07414 \cdot T^2 + 541.5 \cdot e^{-\left( \frac{T}{123.75} \right)^2} \tag{A4}$$





**Table A2.** Constants used in the sublimation calculations for a sphere (addition to Table 2). Where they are different for a hexagonal plate, they are given in Table A4.

| Symbol | Long Name | Value |
|---|---|---|
| $g$ | gravitational constant | $9.81\,\mathrm{m\,s^{-2}}$ |
| $k_\mathrm{T}$ | thermal conductivity of air | $0.024\,\mathrm{J\,m^{-1}\,s^{-1}\,K^{-1}}$ |
| $L_\mathrm{b}$ | lapse rate | $-0.0065\,\mathrm{K\,m^{-1}}$ |
| $M_\mathrm{w}$ | molecular mass of water | $18.02 \times 10^{-3}\,\mathrm{kg\,mol^{-1}}$ |
| $M_\mathrm{air}$ | molecular mass of Earh's air | $28.9644 \times 10^{-3}\,\mathrm{kg\,mol^{-1}}$ |
| $\mu_0$ | viscosity of air at $T = 273\,\mathrm{K}$ and $p = 101325\,\mathrm{Pa}$ (Seinfeld and Pandis, 2006, Table A.7, pg. 1178) | $1.72 \times 10^{-5}\,\mathrm{kg\,m^{-1}\,s^{-1}}$ |
| $p_0$ | reference pressure | $101325\,\mathrm{Pa}$ |
| $R$ | universal gas constant | $8.314\,\mathrm{J\,K^{-1}\,mol^{-1}}$ |
| $R_\mathrm{s}$ | specific gas constant for air | $287.06\,\mathrm{J\,kg^{-1}\,K^{-1}}$ |
| $\rho_\mathrm{i}$ | density of ice | $0.92 \times 10^3\,\mathrm{kg\,m^{-3}}$ |
| $S$ | Sutherland's constant for air (Chapman and Cowling, 1960, Table 15), in a temperature range from $0\,^\circ\mathrm{C}$ to $300\,^\circ\mathrm{C}$ | $114 \pm 24$ |
| $T_0$ | reference temperature | $273.15\,\mathrm{K}$ |
| $T_\mathrm{b}$ | reference temperature in the barometric formula | $288.15\,\mathrm{K}$ |

and the ventilation factor is given by (Pruppacher and Klett, 2010, eq. 13-61):

$$f = 1.0 + 0.108 \cdot \left(\frac{X}{10}\right)^2 \tag{A5}$$

where

$$X = 0.71^{\frac{1}{3}} \cdot N_\mathrm{Re} \tag{A6}$$

$$N_\mathrm{Re} = \frac{2U_\infty r \rho_\mathrm{air}}{\mu} \tag{A7}$$

(Lohmann et al., 2016, eq. 7.36). Where the Reynolds number exceeded the scope of the parameterization, the value for the ventilation factor from the last time step was used. For the terminal velocity, $U_\infty$, $v$ was used. The dynamic viscosity $\mu$ can be derived from Sutherland's formula (Chapman and Cowling, 1960, eq. 12.32-2), which can be rewritten and expanded to:

$$\mu = \frac{BT_0^{\frac{3}{2}}}{S + T_0} + \frac{B\sqrt{T_0}(3S + T_0)(T - T_0)}{2(S + T_0)^2} \tag{A8}$$

with $B = \frac{\mu_0 \cdot (T_0 + S)}{T_0^{\frac{3}{2}}}$.





**Table A3.** Equations used in the sublimation calculations for a hexagonal plate. The other equations used are the same as for a sphere and are given in the text.

| Equation for hexagonal plates | Replaces Eq. |
|---|---|
| $C = 2r/\pi$ (Pruppacher and Klett, 2010, eq. 13-77) | (2) |
| $f = 1.0 - 0.6042 \cdot \left(\frac{X}{10}\right) + 2.79820 \cdot \left(\frac{X}{10}\right)^2 - 0.31933 \cdot \left(\frac{X}{10}\right)^3 - 0.06247 \cdot \left(\frac{X}{10}\right)^4$ where $X = 0.632^{\frac{1}{3}} \cdot N_{\mathrm{Re}}$ (Pruppacher and Klett, 2010, eq. 13-90b) and (Ji and Wang, 1999) | (A5) |
| $m = \rho_i \cdot 9.17 \times 10^{-3} \cdot (2r)^{2.475}$ (Pruppacher and Klett, 2010, Table 2.2a) | (1) and (5) |

**Table A4.** Same as Table 2 but for hexagonal plates. Only those constants that differ from Table 2 are shown.

| Symbol | Long name | Value |
|---|---|---|
| $\alpha$ | coefficient for the velocity-mass-relation for cloud ice (Seifert and Beheng, 2006, Table 1) | $317\,\mathrm{m\,s^{-1}\,kg^{-\beta}}$ |
| $\beta$ | coefficient for the velocity-mass-relation for cloud ice (Seifert and Beheng, 2006, Table 1) | $0.363$ |
| $\gamma$ | coefficient for the velocity-mass-relation for cloud ice (Seifert and Beheng, 2006, Table 1) | $0.5$ |
| $\rho_i$ | density of ice (Pruppacher and Klett, 2010, Table 2.3) | $0.9 \times 10^3\,\mathrm{kg\,m^{-3}}$ |

**Table A5.** Equations used in the sublimation calculations for bullet rosettes. The other equations used are the same as for a sphere and are given in the text.

| Equation for bullet rosettes | Replaces eq. |
|---|---|
| $C = 0.434 \cdot n_{\mathrm{lobes}}^{0.257} \cdot r$ (Chiruta and Wang, 2003) | (2) |
| $f = 1.0 + 0.35463 \cdot \left(\frac{X}{10}\right) + 3.55333 \cdot \left(\frac{X}{10}\right)^2$ where $X = 0.632^{\frac{1}{3}} \cdot N_{\mathrm{Re}}$ (Pruppacher and Klett, 2010, eq. 13-90c) and (Ji and Wang, 1999) | (A5) |
| $m = \alpha_{\mathrm{br}} \cdot (2r \times 10^2)^{\beta_{\mathrm{br}}} \times 10^{-3}$ (Heymsfield and Iaquinta, 2000) | (1) and (5) |
| $\rho_i = 0.78 \cdot (r \cdot 10^3)^{-0.0038} \cdot 10^3 \mathrm{kg\,m^{-3}}$ (Pruppacher and Klett, 2010, Table 2.3) | $\rho_i$ in Table A2 |
| $v = x \cdot (2r \times 10^2)^y \times 10^{-2}$ (Heymsfield and Iaquinta, 2000) | 6 |





**Table A6.** Same as Table 2 but for bullet rosettes. Only those constants that differ from Table 2 are shown.

| Symbol | Long name | Value |
|---|---|---|
| $\alpha_{\mathrm{br}}$ | coefficient for the mass-radius-relation (Heymsfield and Iaquinta, 2000) | $1.25 \times 10^{-5}$ |
| $\beta_{\mathrm{br}}$ | coefficient for the mass-radius-relation (Heymsfield and Iaquinta, 2000) | 1.52 |
| $n_{\mathrm{lobes}}$ | number of lobes in a bullet rosette (typical value, Heymsfield and Iaquinta (2000)) | 3 |
| $x$ | coefficient for the velocity-radius-relation (Heymsfield and Iaquinta, 2000) | 2150 |
| $y$ | coefficient for the velocity-radius-relation (Heymsfield and Iaquinta, 2000) | 1.225 |





## Appendix B: Additional DARDAR analysis

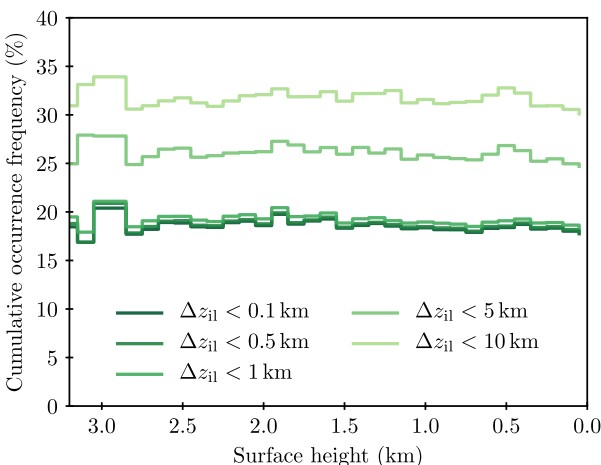

**Figure B1.** Distribution of $\Delta z_{il}$ with underlying surface topography.

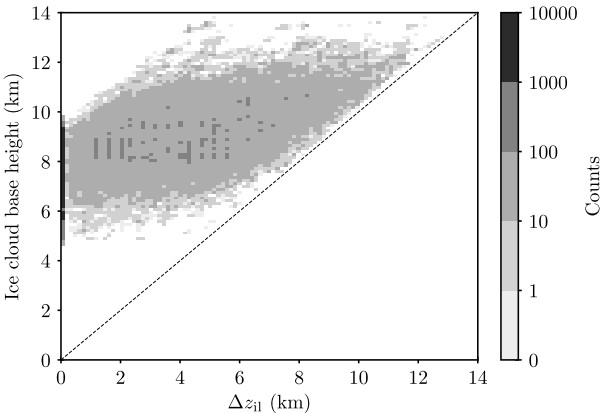

**Figure B2.** Distribution of *icebase* with $\Delta z_{il}$.





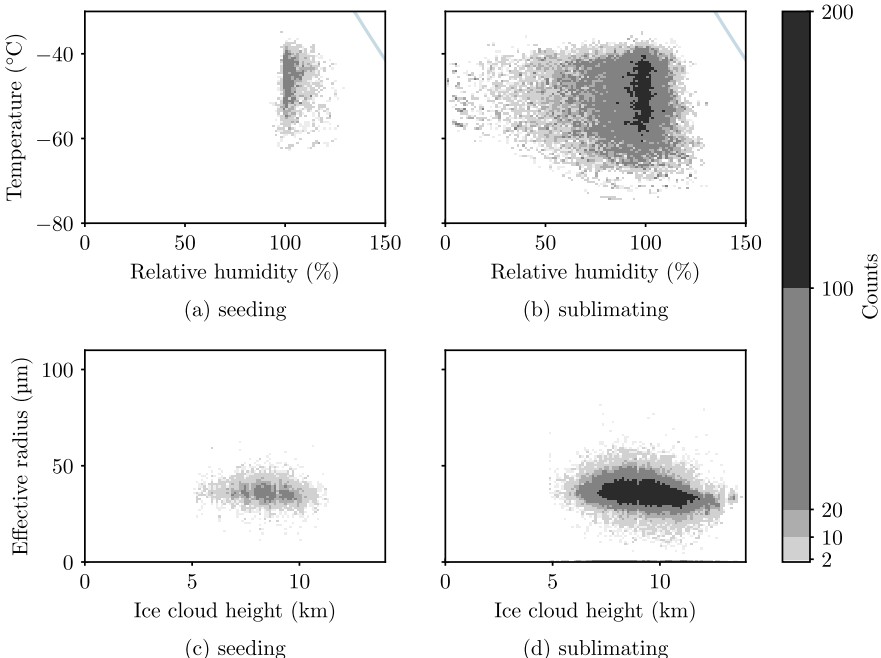

**Figure B3.** Same as Fig. 5, but assuming bullet rosettes as seeding ice crystals.





**Appendix C: Cloud cover data comparison to CALIPSO**

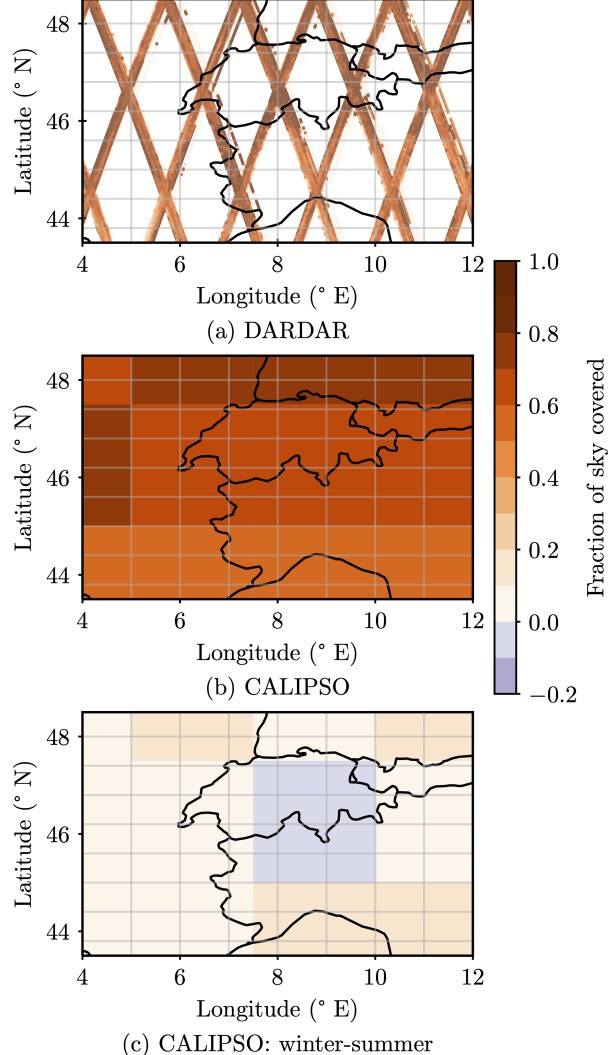

**Figure C1.** Comparison between **(a)** cloud cover derived from the DARDAR satellite product in this study and **(b)** CALIPSO-GOCCP total fraction of sky covered (2006-2017) (Chepfer et al., 2010, 2013). For the DARDAR data, the cloud cover was calculated as the mean (over all tracks within 2006-2017) of the sum of all fractions of sky covered (sum of frac_cov at all temperatures) at each grid point. Sums that were larger than 1 were set to be 1. This method corresponds to the assumption of minimal overlap. **c)** CALIPSO-GOCCP seasonal difference in total cloud cover. To allow for a visual comparison, DARDAR cloud cover data was filtered with a mean over $10 \times 10$ squares.



*Author contributions.* UP conducted the data analysis and sublimation calculations, analysed the results, and was the main author of the
paper. VB developed the initial version of the data analysis and sublimation code. DN conceived the idea of the study. ZD, UL and DN
contributed to the design of the study and the analysis of the results. All authors contributed to the writing of the study.

*Competing interests.* The authors declare no conflict of interest.

*Acknowledgements.* We thank Maiken Vassel for her advice during the development of the sublimation calculations. We thank the AERIS/ICARE
Data and Services Center for providing access to the data used in this study. Sublimation calculations were generated using Copernicus
Climate Change Service Information. Throughout this study, the programming languages CDO (Schulzweida, 2018) and Python (Python
Software Foundation, www.python.org) were used to handle data and analyse it. The satellite data analysis and the sublimation calculations
were conducted with Python as well. This project has received funding from the European Union's Horizon 2020 research and innovation
programme under grant agreement No 821205 (FORCeS).



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
