# Peer review of "How frequent is natural cloud seeding from ice cloud layers (< -35 °C) over Switzerland?"

_Atmospheric Chemistry and Physics, 2020_

## Referee Comment (RC1) · Anonymous Referee #3 · 6 Jan 2021

Review of

**How frequent is natural cloud seeding over Switzerland?**

by Proske et al.

**General comment:**

This study develops a method to investigate occurrence frequencies of natural cloud seeding based on DARDAR satellite products. The region of Switzerland is used here as an example, but it is emphasized that the method can be applied to other areas as well. Two seeding cases are separated in the study, the first represent cirrus above other clouds with the -35∘C isotherm in between, while in the second the cirrus are part of thicker clouds with the -35∘C isotherm inside. Topographical, day/night and seasonal variations of the frequencies of seeding situations are analyzed. Further, sublimation calculations for the seeder ice crystals are performed showing the part that do not sublimate before reaching the lower feeder cloud. In addition, a method to identify in-situ and liquid origin cirrus clouds based on the DARDAR mean effective ice particle size is presented.

The topic of the paper is well suited for ACP and also is of high scientific interest. The methods used are scientifically sound and the results are robust and provide new insights into the field. Thus I recommend the paper for publication in ACP with minor changes. Below you find a number of comments/recommendations to consider for the final version of the paper.

**Specific comments:**

**1) Figure 1** (and elsewhere in the text): I guess that the indices ‚i‘ and ‚l‘ in Δz$_{il}$ mean ‚ice‘ and ‚liquid‘. To my opinion it would be more consistent to change the ‚l‘ to ‚m‘, because the corresponding clouds are termed mixed-phase clouds and could be ice, liquid or mixed.

**2) Page 2, line 50 ff:** You might take into account to add Wolf et al. (2018), ACP, to the listed references (here and later). They sorted ice particle shapes and size distributions according to liquid and in-situ origin cirrus clouds.

Wolf, V., Kuhn, T., Milz, M., Voelger, P., Krämer, M., and Rolf, C.: Arctic ice clouds over northern Sweden: microphysical properties studied with the Balloon-borne Ice Cloud particle Imager B-ICI, Atmos. Chem. Phys., 18, 17371–17386, https://doi.org/10.5194/acp-18-17371-2018, 2018.

**3) Page 2, lines 54:** ‚ … *ice can only be formed via heterogeneous nucleation on ice nucleating particles (Kanji et al., 2017).*‘

I suggest to change this to ‚ … *ice can only be formed via heterogeneous nucleation on ice nucleating particles (e.g. Kanji et al., 2017, and references therein).*‘ because this has been known for a long time and Kanji et al. is a recent overview paper.

**4) Page 2, line 56 ff:** ‚ … *the Wegener-Bergeron-Findeisen process, where ice crystals grow at the expense of liquid droplets, when the saturation ratio is between saturation with respect to water and ice ...*‘

The WBF process is where the water vapor saturation ratio is between subsaturation with respect to water and supersaturation with respect to ice ($S_w < 1$ and $S_i > 1$).

**5) Page 3, line 58 ff:**  You might consider to add the recent article of  Korolev and Leisner (2020), ACP,  to the references of secondary ice production:

Korolev, A. and Leisner, T.: Review of experimental studies of secondary ice production, Atmos. Chem. Phys., 20, 11767–11797, https://doi.org/10.5194/acp-20-11767-2020, 2020.

**6) Page 5, line 101 ff:** *‚The two satellites are designed for their data to be combined:  the lidar on CALIPSO is able to identify the thin upper layers of cirrus clouds that the radar on CloudSat misses (Winker et al., 2010), while the latter is able to look through thick clouds where the lidar beam is attenuated.'*

You mention the lidar (thin) and radar (thick) clouds, but where/what kind  are the  clouds from the visible camera and a three-channel infrared radiometer noted before? If all instruments are combined, does the DARDAR product cover the whole range of  clouds or are the thinnest/thickest missed ?  Can you give an estimate here on the percentag eof missed clouds ? This might be important in the especially for thin cirrus, yes ?

**7) Page 5, line 119-120:** *‚…  DARDAR categories 1, 2, 3 and 4 (ice, ice + supercooled, liquid> −35◦C and supercooled) …'*  These are only 3 categories,  liquid > −35 ◦ C and supercooled are the same.

**8) Page 6, line 124-126:** *‚…  liquid cloud droplets have been found to supercool to −35 ◦ C before freezing homogeneously (Murray et al., 2010; Herbert et al., 2015). The temperature for homogeneous freezing of water droplets is also often given as −38 ◦C (Kanji et al., 2017).'*

There are earlier references for the existence of supercooled drops and also for the  temperature of homogeneous drop freezing  … e.g. Pruppacher & Klett ?

**9) Page 7, line 130:**  *‚…  air density, air temperature and the relative humidity determine the ice crystal sublimation rate and fall velocity.'*

Aren't the the ice crystal size  and the vertical velocity of the air  also important for the fall velocity?  In line 143 you metion that you use reff...

**10) Page 7, line 141-142:** *‚Relative humidity and temperature were therefore taken from ERA5 reanalysis data...'*

What about the quality of ERA5 RH in the  cirrus temperature range? Isn't there a dry bias ?

**11) Page 9, a) line 190-196:** The percentages stated here, the numbers in the caption of  Figure 3 and the graphs shown in  Figure 3 b seems not to be consistent. Maybe I misunderstand something, but then it would be good to better explain.

 **b)  line 197:**  *‚In 56 % of these cases (18 % in total), $\Delta z_{il}$ is smaller than 100 m.'*

Shouldn't these numbers be visible in  Figure 3 b (see also previous comment) ? I see the 18 % for cloud free conditions, but the percentage of cirrus clouds with $\Delta z_{il}$  smaller than 100 m is ~41 %, not 56 % ?

**12) Page 11, line 219:** ‚*Our results for multi-layer cloud occurrence frequency are similar to but smaller than the ones given in the literature.*‘

‚Similar to but smaller than‘ is hardly possible …
Also, for convenience for the reader, please repeat the number of ‚our results for multi-layer cloud occurrence frequency‘ so that there is a directly comparison of the numbers in the text.

**13) Table 3:** The sum of the first two numbers in the first row (18 + 13 = 31 %, all seasons, all day, whole domain) should be the same as stated on page 9, line 194, yes ?
‚*32 % of the measurements contain both a cirrus and a mixed-phase cloud simultaneously.*‘

This number could be repeated for the convenice of the readers.

**14) Page 11, line 248:** ‚*Table 3 also contains the results of a climatological analysis of Δzil.*‘

I recommend to say ‚seasonal‘ instaed of ‚climatological‘.

**15) Page 11, line 249:** ‚*The relative increase is similar …*‘

More clear is: The relative increase of the fractions of Δzil is similar …

**16) Page 12, line 250-252:** ‚*There is no noticeable difference in frequencies during day and night.*‘

There are noticeable  differences between day and night (winter), and also differences between the summer and winter  days and nights.    Please clarify.

**17) Page 12, line 266:** You might cite here also Krämer et al. (2000), ACP.

Krämer, M., Rolf, C., Spelten, N., Afchine, A., Fahey, D., Jensen, E., Khaykin, S., Kuhn, T., Lawson, P., Lykov, A., Pan, L. L., Riese, M., Rollins, A., Stroh, F., Thornberry, T., Wolf, V., Woods, S., Spichtinger, P., Quaas, J., and Sourdeval, O.: A microphysics guide to cirrus – Part 2: Climatologies of clouds and humidity from observations, Atmos. Chem. Phys., 20, 12569–12608, https://doi.org/10.5194/acp-20-12569-2020, 2020.

**18) Page 13, line 276-277:** ‚ *… freeze  … predominantly homogeneously at temperatures below−35◦C.*‘

More correct:  ‚… at temperatures around −35 ◦ C.‘

**19) Page 13, line 296 ff:** ‚*This suggests that the influence of the liquid origin on the microphysical properties of the cirrus clouds is lost once the clouds are lifted, for example because the large ice crystals sediment out, or that lifting of entire clouds above the −35 ◦C isotherm hardly ever occurs. Wernli et al. (2016), who investigated the frequency of the formation pathways in a trajectory-based analysis, already noted that ice crystal sedimentation and cloud turbulence could "potentially alter the local cirrus characteristics and 'confuse' the simple categorization". This seems to be the case with the data presented here, or otherwise the data suggests that liquid origin clouds are hardly ever lifted entirely above the −35 ◦ C isotherm.*‘

This is a very good and sound discussion and Figure 4 provides new insights in the characteristics of liquid and in-situ origin cirrus.  I like to add here that I think  that the latter suggestion ‚*liquid origin clouds are hardly ever lifted entirely* above the −35 ◦ C isotherm.‘ is more likely. It is true

that especially ice crystal sedimentation alter the characteristics of liquid origin clouds, but not to such an extent that it completely disappears, but the influence decreases with increasing altitude, because the altitude the ice crystals reach depends on their size and the updraft, i.e. the higher the altitude the smaller the largest ice crystals  (see also Luebke et al. 2016  and Krämer et al., 2020). But such an effect is nearly not visible in Figure 4c. Another argument that liquid origin clouds are hardly ever lifted entirely above the $-35°$C isotherm is that they appear mostly in meteorological systems with large vertical extents, namely in warm conveyor belts or convection.

**20) Page 16, line 325: ,***Figure 5c shows that ice crystals do not survive the fall from cirrus cloud base heights above 11 km.'*

… which correpsonds to the temperature limit of $-65°$C from Figure 5a.

**21) Page 18, line 377-378:** *,... where they act as seeds for the glaciation of clouds.'*  … in case they fall in an environment  that is subsaturated with respect to water – otherwise (supersaturated) they would grow.
By the way: did you consider the updraft in the sublimation calculations? Maybe a point to mention at the appropriate place in the paper.

**22) Page 18, line 390: ,***In sublimations calculations ...'*   Remove the latter ,s' in  sublimations.

---

## Referee Comment (RC2) · Anonymous Referee #2 · 11 Jan 2021

The paper is well written, contains original and interesting results, and is therefore appropriate for ACP.

Some general remarks in the beginning. The paper (title) deals with a good question! But do we get a proper answer? Seeding of liquid-water clouds by cirrus is one branch, what about another, certainly relevant pathway via stratiform mixed-phase clouds?

With other words: The impact of cirrus on cloud seeding . . . is in the focus of the article. Ok! However, what about all the mid-tropospheric stratiform clouds (altocumulus, stratus, stratocumulus)? Ice crystals may form via immersion freezing mode. These crystals grow fast in the liquid-water environment, fall out and enter the next liquid-water cloud layer and produce large amounts of ice. This aspect is not covered by the paper, but should be discussed to give the reader a clear chance to judge the value of

your paper.

Then I struggled with this classification: in situ origin cirrus. . . . and liquid origin cirrus!

In former times, there was a clear separation between outflow cirrus (convectively generated cirrus, as remnants of big cumulus towers. . .) and synoptic cirrus (cirrus uncinus, cirrocumulus, cirrostratus). And it was clear that outflow cirrus must have quite different properties than cirrus that formed homogeneously or heterogeneously via deposition nucleation.

'In situ' is some kind of a property and used to describe in situ observations, in situ experiments, in situ instruments, especially to contrast them from remote observation and remote sensing instruments. . .. But what about 'in situ origin' cirrus? . . . I know what you want to say, but is that a proper designation? Is that even correct English?

Locally generated cirrus vs convectively generated cirrus would sound better.

Liquid origin: 'liquid' is not precise . . .. could be even sulfate aerosol droplets or oil droplets.

One should discuss this kind of designation in a broader way to corroborate that this kind of classification of cirrus is open for discussion.

Some details:

Introduction:

P2: It took me some time after reading all the complex aspects given in the introduction what the goal of the study is: We concentrate on the cirrus impact only! I would suggest to simply state what the seeder-feeder mechanism is (definition), what kind (branches) of seeder-feeder pathways exist, and that you want to concentrate on the one via pure cirrus. . ., and why you restrict your study to this specific field (because only for this one can use DARDAR. . ., if I understood correctly). That would be more simple and straight forward.

P3: Figure 3. Case (c), the right panel is confusing. The orange line indicates -35°C (?) and then you have ice (100%) above the respective height (at temperatures below -35°C)? . . . and liquid water droplets (100%) below this height? Exactly for all temperatures higher than -35°C? Is that realistic? Maybe in the case of a thunderstorm with 30m/s upwind. . ... that may be the case, i.e., only water below the orange line. But with slow updrafts and ice sedimentation the picture is certainly more complicated.

P9, L193: Please explain in more detail: You found scences with cirrus and liquid-water cloud in 32% out of all cases, and then, in 77% out of these 32% cases, a liquid cloud directly below the cirrus? . . .so that the seeder-feeder process can work?

Question: How do you know that the liquid-water cloud is free of ice crystals? Because of the radar observations? Please explain!

P13, L270: I am sure that there are papers from the 1980s-1990s distinguishing the microphysical properties of outflow and synoptic cirrus. Please check! Or did this kind of research started just a few years ago as your references indicate?

P13, L273: By listing all available mechanisms, step by step, starting from temperatures above -35°C, and then going to temperatures below -35°C at which both homogenous and heterogenbous ice nucleation can occur. . ... the separation into different cirrus classes would be process-based and more easy to understand. Why do you not mention the immersion freezing process?

P13, L285: What do you want to say? : It also confirms . . . that liquid origin cirrus clouds are composed of ice crystals. . ...I mean: a white horse is white. . . otherwise it is not a white horse.

P13, I find the full discussion on pager 13 quite a bit too complex and many times confusing.

P18: My question remains: Would be nice to have some speculation (some estimation, your opinion) on the relative impact of stratiform clouds (mixed-phase altocumulus

etc. . .) on the seeder-feeder processes.

All in all: A good paper!
* * *

---

## Referee Comment (RC3) · Anonymous Referee #1 · 14 Jan 2021

In their paper "How frequent is natural cloud seeding over Switzerland", the authors quantify the occurrence of an ice cloud layer above other clouds from CALIOP/CloudSat (DARDAR) data. In addition, the possibility of ice crystals sedimenting into the lower layers is calculated from temperature and relative humidity profiles based on ERA5.

The paper is well written, the method is sound, the analysis is careful and the results are very interesting. I recommend the manuscript for publication after minor revisions.

Main comments:

- I don't understand the "filtering" or smoothing of the data. Line 121 mentions a 7x7x7 points cube, but what are these three dimensions? Why is the median used? How

sensitive are the results to this filtering?

- Line 131 says that reff is also filtered "for consistency", but the smoothing applied here is very different. What impact does this have on the results?

- The temperature in the DARDAR dataset is also not a retrieved parameter, but is obtained from interpolated NWP data, if I'm not mistaken. How different is this from the ERA5 temperature? Is the discrepancy in the sublimation calculations only due to a discrepancy in relative humidity or also in temperature? How is this affected by the smoothing?

- Sometimes the terminology is a little unconventional. The term "seeder-feeder" is also used for cases in which ice crystals fall into pure ice clouds. I don't think the term is appropriate in this case. Further, the ice cloud layer below -35°C is termed "cirrus", even if it is the upper part of a mixed phase cloud – e. g. a frontal cloud or a convective cloud. I would reserve "cirrus" for isolated ice clouds.

Other minor comments:

- Table 2 gives coefficients "for cloud droplets", shouldn't this be ice particles?

- Fig. 3(b): Why does the cumulative occurrence frequency for the situation with a cirrus cloud does not reach 1 minus the cirrus cloud frequency (i.e. a little less than 0.6) at 10km?

- What is the sensitivity to the time step of the sublimation calculations?

- Line 140: "mean values" of what?

- Line 416: I suggest to include the reference to the companion paper only if it is already available (at least as preprint) when the revised version of this manuscript is published. Otherwise, this is more confusing than helpful for the readers.

Technical comments:

- Caption of Fig. 3: typo "atmposphere"

- Line 352: "for spheres": blanks missing.

- Table A2: typo "Earh"

---

## Author Comment (AC1) · 23 Feb 2021

**Author Response to Reviews of**

**How frequent is natural cloud seeding over Switzerland?**

Ulrike Proske, Verena Bessenbacher, Zane Dedekind, Ulrike Lohmann, and David Neubauer
*Atmospheric Chemistry and Physics,* `doi:10.5194/acp-2020-1145`
* * *
RC: *Reviewer Comment*,     AR: *Author Response*,     ☐ Manuscript text

We sincerely thank the reviewers for their thoughtful and constructive feedback. We implemented their feedback into a revised version of the manuscript. Please find our answer to the reviewers' points below, followed by a marked-up manuscript verison.

**1. Reviewer Comment #1**

RC: *This study develops a method to investigate occurrence frequencies of natural cloud seeding based on DARDAR satellite products. The region of Switzerland is used here as an example, but it is emphasized that the method can be applied to other areas as well. Two seeding cases are separated in the study, the first represent cirrus above other clouds with the $-35\,°C$ isotherm in between, while in the second the cirrus are part of thicker clouds with the $-35\,°C$ isotherm inside. Topographical, day/night and seasonal variations of the frequencies of seeding situations are analyzed. Further, sublimation calculations for the seeder ice crystals are performed showing the part that do not sublimate before reaching the lower feeder cloud. In addition, a method to identify in-situ and liquid origin cirrus clouds based on the DARDAR mean effective ice particle size is presented. The topic of the paper is well suited for ACP and also is of high scientific interest. The methods used are scientifically sound and the results are robust and provide new insights into the field. Thus I recommend the paper for publication in ACP with minor changes. Below you find a number of comments/recommendations to consider for the final version of the paper.*

AR: *Thank you for your clear and detailed feedback. Please find our respective answers directly below your comments below.*

**1.1. Figure 1 (and elsewhere in the text)**

RC: *I guess that the indices ‚i‘ and ‚l‘ in $\Delta z_{il}$ mean ‚ice‘ and ‚liquid‘. To my opinion it would be more consistent to change the ‚l‘ to ‚m‘, because the corresponding clouds are termed mixed-phase clouds and could be ice, liquid or mixed.*

AR: *We agree and changed $\Delta z_{\mathrm{il}}$ to $\Delta z_{\mathrm{im}}$ everywhere for consistency.*

**1.2. Page 2, line 50 ff**

RC: *You might take into account to add Wolf et al. (2018), ACP, to the listed references (here and later). They sorted ice particle shapes and size distributions according to liquid and in-situ origin cirrus clouds.*
*Wolf, V., Kuhn, T., Milz, M., Voelger, P., Krämer, M., and Rolf, C.: Arctic ice clouds over northern Sweden: microphysical properties studied with the Balloon-borne Ice Cloud particle Imager B-ICI, Atmos. Chem. Phys., 18, 17371–17386, https://doi.org/10.5194/acp-18-17371-2018, 2018.*

AR:     *We added this reference in line 51, 52, 273, and 389 of the original manuscript version.*

**1.3.    Page 2, lines 54**

RC:     **, ... ice can only be formed via heterogeneous nucleation on ice nucleating particles (Kanji et al., 2017).' I suggest to change this to , ... ice can only be formed via heterogeneous nucleation on ice nucleating particles (e.g. Kanji et al., 2017, and references therein).' because this has been known for a long time and Kanji et al. is a recent overview paper.**

AR:     *You are right. We changed this as you suggested:*

> Seeding ice crystals can have a large influence on cloud properties, because in the atmosphere, at temperatures warmer than $-38\,^\circ$C, ice can only be formed via heterogeneous nucleation on ice nucleating particles (Kanji et al., 2017).(e.g. Kanji et al. (2017), and references therein).

**1.4.    Page 2, line 56 ff**

RC:     **, ... the Wegener-Bergeron-Findeisen process, where ice crystals grow at the expense of liquid droplets, when the saturation ratio is between saturation with respect to water and ice ...' The WBF process is where the water vapor saturation ratio is between subsaturation with respect to water and supersaturation with respect to ice (Sw < 1 and Si > 1).**

AR:     *We changed the wording as follows to make the required conditions more clear:*

> *Once ice particles are formed within the cloud or enter the cloud from outside, they grow by riming or vapour deposition (rapidly via the Wegener-Bergeron-Findeisen process, where ice crystals grow at the expense of liquid droplets, when the water vapor saturation ratio is between saturation subsaturated with respect to water and supersaturated with respect to ice (Wegener, 1911; Bergeron, 1935; Findeisen, 1938)) . . .*

**1.5.    Page 3, line 58 ff**

RC:     **You might consider to add the recent article of Korolev and Leisner (2020), ACP, to the references of secondary ice production:**
        **Korolev, A. and Leisner, T.: Review of experimental studies of secondary ice production, Atmos. Chem. Phys., 20, 11767–11797, https://doi.org/10.5194/acp-20-11767-2020, 2020.**

AR:     *Thank you for pointing us to that work. We added a reference as follows:*

> *. . . and can multiply through secondary ice production (Korolev and Leisner (2020), Hallett-Mossop process (Hallett and Mossop, 1974; Mossop and Hallett, 1974; Mossop and Hallett, 1974), frozen droplet shattering (Lauber et al., 2018), or ice-ice collisional breakup (Sullivan et al., 2018)).*

**1.6.    Page 5, line 101 ff**

RC:     **,The two satellites are designed for their data to be combined: the lidar on CALIPSO is able to identify the thin upper layers of cirrus clouds that the radar on CloudSat misses (Winker et al., 2010), while the**

*latter is able to look through thick clouds where the lidar beam is attenuated.' You mention the lidar (thin) and radar (thick) clouds, but where/what kind are the clouds from the visible camera and a three-channel infrared radiometer noted before? If all instruments are combined, does the DARDAR product cover the whole range of clouds or are the thinnest/thickest missed ? Can you give an estimate here on the percentage of missed clouds ? This might be important in the especially for thin cirrus, yes ?*

AR: *The visible camera and the three-channel infrared radiometer are mentioned here to complete the list of instruments onboard CALIPSO. DARDAR-cloud only uses the radar data from CloudSat and the lidar data from CAlIPSO. Delanoë and Hogan (2010b) also discuss the possibility to combine the radar and lidar data with infrared radiometer data from the A-train satellite MODIS, but this product (DARDAR-rad-cloud) is not available yet.*

> The  DARDAR-CLOUD satellite data product used in this study is based on radar  and lidar data from the CloudSat and CALIPSO satellites.

*Radar is less sensitive to small particles, while lidar suffers from attenuation Delanoë and Hogan (2010b). DARDAR CLOUD retrieves clouds detected by both or just one of the instruments. Krämer et al. (2020) mention "a decreasing detectability of thin cirrus at $T \leq 190\,\mathrm{K}$" in the DARDAR-$N_{\mathrm{ice}}$ climatology. Also, CloudSat and CALIPSO have been found to miss clouds near the surface, within the lowest $1\,\mathrm{km}$ (e.g. Chan and Comiso (2011) and Liu et al. (2015)). However, these clouds are not relevant for this study, because Krämer et al. (2020) discuss tropical clouds and therefore their $\Delta z_{\mathrm{il}}$ is too large. We added this statement to the manuscript:*

> Cirrus clouds in the tropical tropopause layer and clouds close to the surface are known to be missed by the radar and lidar on CloudSat and CALIPSO (Chan and Comiso, 2011; Liu et al., 2015; Krämer et al., 2020), but these are not relevant for this study.

**1.7. Page 5, line 119-120**

RC: *,... DARDAR categories 1, 2, 3 and 4 (ice, ice + supercooled, liquid>$-35\,°\mathrm{C}$ and supercooled) ...' These are only 3 categories, liquid > $-35\,°\mathrm{C}$ and supercooled are the same.*

AR: *You are correct. The DARDAR cloud product specifies the four categories 1, 2, 3 and 4 as ice, ice + supercooled, liquid warm and supercooled (Delanoë and Hogan, 2010a). The disctinction between the two latter categories is not solely based on temperature, but also on pressure and humidity as explained in Delanoë and Hogan (2010b).*

> All variables were derived from a cloud mask, where the DARDAR categories 1, 2, 3 and 4 (ice, ice + supercooled, liquid  (warm and supercooled)) were combined to simply signify the presence of cloud layers.

**1.8. Page 6, line 124-126**

RC: *,... liquid cloud droplets have been found to supercool to $-35\,°\mathrm{C}$ before freezing homogeneously (Murray et al., 2010; Herbert et al., 2015). The temperature for homogeneous freezing of water droplets is also*

*often given as* $-38\,°\mathrm{C}$ *(Kanji et al., 2017).' There are earlier references for the existence of supercooled drops and also for the temperature of homogeneous drop freezing ... e.g. Pruppacher & Klett ?*

AR:  *We agree and changed this as you suggested:*

> In this study, cirrus clouds are defined as clouds at temperatures lower than $-35\,°\mathrm{C}$, and mixed-phase clouds are defined as all clouds  with temperatures warmer than $-35\,°\mathrm{C}$.  Depending on their size, liquid cloud droplets  supercool to $-35\,°\mathrm{C}$ to $-40\,°\mathrm{C}$ before freezing homogeneously   (e.g. Pruppacher and Klett (2010), Murray et al. (2010), Herbert et al. (2015), and Kanji et al. (2017)).

**1.9. Page 7, line 130**

RC:  *,... air density, air temperature and the relative humidity determine the ice crystal sublimation rate and fall velocity.' Aren't the the ice crystal size and the vertical velocity of the air also important for the fall velocity? In line 143 you metion that you use reff...*

AR:  *We have changed the wording here to indicate that the environmental parameters are only a part of what determines the sublimation rate and fall velocity:*

> *Environmental parameters such as the air density, air temperature and the relative humidity  also affect the ice crystal sublimation rate and fall velocity.*

**1.10. Page 7, line 141-142**

RC:  *,Relative humidity and temperature were therefore taken from ERA5 reanalysis data...' What about the quality of ERA5 RH in the cirrus temperature range? Isn't there a dry bias ?*

AR:  *We are not aware of a dry bias in ERA5. Hoffmann et al. (2019) mention a stratospheric dry bias in ERA interim that was reduced for ERA5, which is not relevant for this study. Earlier ECMWF reanalysises have been found to have mid tropospheric dry bias, at least in the tropics (Lohmann et al., 1995), but since we found no other mention of this dry bias for ERA5, we have reason to believe that it is outdated.*

**1.11. Page 9, a) line 190-196**

RC:  *The percentages stated here, the numbers in the caption of Figure 3 and the graphs shown in Figure 3 b seems not to be consistent. Maybe I misunderstand something, but then it would be good to better explain.*

AR:  *Thank you for noticing this mistake! Indeed, the total number of measurements is 853833, with 267354 measuring $\Delta z_{\mathrm{im}}$ and 355331 measuring cirrus clouds. The corresponding line in the caption now reads as follows:*

> *The total number of measurements is 853833, with  267354 measuring  $\Delta z_{\mathrm{im}}$ and 355331 measuring cirrus clouds.*

*Consequently, the corresponding percentages change to:*

> 31 % of the measurements contain both a cirrus and a mixed-phase cloud simultaneously. . . . *Tailoring this result to the sedimentation of ice crystals from a cirrus cloud,* 75 % *of the measurements that detect a cirrus cloud also detect a lower mixed-phase cloud.*

**1.12. line 197**

**RC:** *,In 56 % of these cases (18 % in total), $\Delta z_{il}$ is smaller than $100\,\mathrm{m}$.' Shouldn't these numbers be visible in Figure 3 b (see also previous comment) ? I see the 18 % for cloud free conditions, but the percentage of cirrus clouds with $\Delta z_{il}$ smaller than $100\,\mathrm{m}$ is $\approx$ 41 %, not 56 % ?*

**AR:** *This also is a mistake. 44% is the correct fraction of $\Delta z_{il}$ smaller than $100\,\mathrm{m}$ in cases with cirrus clouds present (instead of 56%). The text was corrected as follows:*

> *In*  44 % of the cases with a detected cirrus cloud (18 % *in total),* $\Delta z_{im}$ *is smaller than* $100\,\mathrm{m}$.

**1.13. Page 11, line 219**

**RC:** *,Our results for multi-layer cloud occurrence frequency are similar to but smaller than the ones given in the literature.' ,Similar to but smaller than' is hardly possible ... Also, for convenience for the reader, please repeat the number of ,our results for multi-layer cloud occurrence frequency' so that there is a directly comparison of the numbers in the text.*

**AR:** *We agree with your propositions and incorporated them as follows:*

> *Our results for multi-layer cloud occurrence frequency* , 13 % ($\Delta z_{im} > 100\,\mathrm{m}$), are *smaller than the ones given in the* following *literature.*

**1.14. Table 3**

**RC:** *The sum of the first two numbers in the first row (18 + 13 = 31 %, all seasons, all day, whole domain) should be the same as stated on page 9, line 194, yes ? ,32 % of the measurements contain both a cirrus and a mixed-phase cloud simultaneously.' This number could be repeated for the convenice of the readers.*

**AR:** *With the changes applied in comment 1.11 the two statements agree now.*
*Also, we added a sentence reminding the reader of the total fraction of $\Delta z_{im}$ to the table caption.*

> In total, 31 % of the measurements contain both a cirrus and a mixed-phase cloud. $\Delta z_{im}$ *up to* $12\,\mathrm{km}$ *in length were evaluated.*

**1.15. Page 11, line 248**

**RC:** *,Table 3 also contains the results of a climatological analysis of $\Delta z_{il}$.' I recommend to say ,seasonal' instaed of ,climatological'.*

**AR:** *We agree and changed this accordingly in the text and in the table.*

> *Table 3 also contains the results of  the seasonal analysis of $\Delta z_{\text{im}}$.*

**1.16. Page 11, line 249**

**RC:** *‚The relative increase is similar ...' More clear is: The relative increase of the fractions of $\Delta z_{il}$ is similar ...*

 **AR:** *We agree and changed this accordingly.*

> *The relative increase of the fractions of $\Delta z_{\text{im}}$ is similar for the smaller ($\Delta z_{\text{im}} < 100\,\text{m}$) and the larger distances ($\Delta z_{\text{im}} > 100\,\text{m}$).*

**1.17. Page 12, line 250-252**

**RC:** *There is no noticeable difference in frequencies during day and night.' There are noticeable differences between day and night (winter), and also differences between the summer and winter days and nights. Please clarify.*

 **AR:** *We changed this statement to be more specific as follows:*

>  Other than the increase in $\Delta z_{\text{im}}$ during night in winter, there are no substantial differences in frequencies between day and night within a seasonal category.

**1.18. Page 12, line 266**

**RC:** *You might cite here also Krämer et al. (2000), ACP. Krämer, M., Rolf, C., Spelten, N., Afchine, A., Fahey, D., Jensen, E., Khaykin, S., Kuhn, T., Lawson, P., Lykov, A., Pan, L. L., Riese, M., Rollins, A., Stroh, F., Thornberry, T., Wolf, V., Woods, S., Spichtinger, P., Quaas, J., and Sourdeval, O.: A microphysics guide to cirrus – Part 2: Climatologies of clouds and humidity from observations, Atmos. Chem. Phys., 20, 12569–12608, https://doi.org/10.5194/acp-20-12569-2020, 2020.*

 **AR:** *Thanks for pointing us to this reference. We added it as suggested.*

**1.19. Page 13, line 276-277**

**RC:** *, ... freeze ... predominantly homogeneously at temperatures below $-35\,°\text{C}$.' More correct: '... at temperatures around $-35\,°\text{C}$.'*

 **AR:** *We agree and changed this accordingly.*

> *These studies distinguish in situ origin cirrus clouds, which form by homogeneous nucleation of solution droplets or heterogenous nucleation of ice nucleating particles within the cirrus temperature range, and liquid origin clouds, which form from supercooled water droplets which are uplifted to the cirrus temperature range and freeze either heterogeneously at warmer temperatures or predominantly homogeneously at temperatures  around $-35\,°\text{C}$.*

**1.20.  Page 13, line 296 ff**

**RC:** *‚This suggests that the influence of the liquid origin on the microphysical properties of the cirrus clouds is lost once the clouds are lifted, for example because the large ice crystals sediment out, or that lifting of entire clouds above the $-35\,°C$ isotherm hardly ever occurs. Wernli et al. (2016), who investigated the frequency of the formation pathways in a trajectory-based analysis, already noted that ice crystal sedimentation and cloud turbulence could "potentially alter the local cirrus characteristics and 'confuse' the simple categorization". This seems to be the case with the data presented here, or otherwise the data suggests that liquid origin clouds are hardly ever lifted entirely above the $-35\,°C$ isotherm.' This is a very good and sound discussion and Figure 4 provides new insights in the characteristics of liquid and in-situ origin cirrus. I like to add here that I think that the latter suggestion ‚liquid origin clouds are hardly ever lifted entirely above the $-35\,°C$ isotherm.' is more likely. It is truethat especially ice crystal sedimentation alter the characteristics of liquid origin clouds, but not to such an extent that it completely disappears, but the influence decreases with increasing altitude, because the altitude the ice crystals reach depends on their size and the updraft, i.e. the higher the altitude the smaller the largest ice crystals (see also Luebke et al. 2016 and Krämer et al., 2020). But such an effect is nearly not visible in Figure 4c. Another argument that liquid origin clouds are hardly ever lifted entirely above the $-35\,°C$ isotherm is that they appear mostly in meteorological systems with large vertical extents, namely in warm conveyor belts or convection.*

**AR:** *Thanks for sharing your ideas! We agree that the distinctively different ice crystal size distributions for the two modes suggest that liquid origin clouds are not altered by sedimentation so much that they are confused for in-situ clouds. We incorporated this point and your thought about the vertical extent as follows:*

> Wernli et al. (2016), who investigated the frequency of the formation pathways in a trajectory-based analysis,  noted that ice crystal sedimentation and cloud turbulence could "potentially alter the local cirrus characteristics and 'confuse' the simple categorization".  However, the distinctively different ice crystal size distributions for the two modes in Fig. 4b) and c) suggest that liquid origin clouds are not altered by sedimentation so much that they are confused for in-situ clouds. Instead, the data suggests that liquid origin clouds are hardly ever lifted entirely above the $-35\,°C$ isotherm, which is likely because of their large vertical extent.

**1.21.  Page 16, line 325**

**RC:** *‚Figure 5c shows that ice crystals do not survive the fall from cirrus cloud base heights above 11 km.' ... which correpsonds to the temperature limit of $-65\,°C$ from Figure 5a.*

**AR:** *We agree and added a clarifying sentence.*

> *We attribute this to smaller ice crystals at these colder temperatures and to the fact that high cirrus cloud bases correspond to large distances to lower lying mixed-phase clouds that ice crystals are less likely to survive. This also explains why ice crystals starting their sedimentation at colder temperatures sublimate more often before reaching a lower cloud than those sedimenting from warmer cloud bases, as the temperature limit of $-65\,°C$ corresponds to the height limit of $11\,km$ (see Figure 5a).*

**1.22. Page 18, line 377-378**

**RC:** *‚... where they act as seeds for the glaciation of clouds.' ... in case they fall in an environment that is subsaturated with respect to water – otherwise (supersaturated) they would grow. By the way: did you consider the updraft in the sublimation calculations? Maybe a point to mention at the appropriate place in the paper.*

**AR:** *Yes, of course the sedimenting ice crystals themselves may also grow within the cloud, but would still initiate glaciation. We have rephrased the sentence as follows:*

> *The seeder-feeder mechanism here refers to ice crystals that fall from a cirrus cloud into a lower cloud, where they  initiate the glaciation of clouds.*

We have not considered updraft in the sublimation calculations and have added a clarifying sentence in the description of the sublimation calculations:

> In between cloud layers, small up- or downdrafts can be expected. For lack of reliable data on such small scales, the updraft velocity was not considered in the sublimation calculations.

The updraft between cloud layers is not available from observations. We expect small scale up- or downdrafts (due to the sublimation of hydrometeors), which are not resolved in ERA5.

**1.23. Page 18, line 390**

**RC:** *‚In sublimations calculations ...' Remove the latter ‚s' in sublimations.*

**AR:** *Thank you for noticing. We changed this as suggested.*

> *In  sublimation calculations we found that a significant number of ice crystals reached the lower cloud layers.*

**2. Reviewer Comment #2**

**RC:** *The paper is well written, contains original and interesting results, and is therefore appropriate for ACP.*

**AR:** *Thank you for your thoughtful feedback. Please find our respective answers directly below your comments below.*

**RC:** *Some general remarks in the beginning. The paper (title) deals with a good question! But do we get a proper answer? Seeding of liquid-water clouds by cirrus is one branch, what about another, certainly relevant pathway via stratiform mixed-phase clouds? With other words: The impact of cirrus on cloud seeding ... is in the focus of the article. Ok! However, what about all the mid-tropospheric stratiform clouds (altocumulus, stratus, stratocumulus)? Ice crystals may form via immersion freezing mode. These crystals grow fast in the liquid-water environment, fall out and enter the next liquid-water cloud layer and produce large amounts of ice. This aspect is not covered by the paper, but should be discussed to give the reader a clear chance to judge the value of your paper.*

AR: *Yes, of course other ice containing clouds may act as seeding clouds as well. We had briefly mentioned that in the Conclusion: "This study focuses on natural cloud seeding that originate from cirrus clouds, but seeding ice crystals can also sediment from mixed-phase clouds." For clarity, we have now added a sentence to the Introduction as well:*

> *This study focuses on cirrus clouds as seeding clouds because they can be identified readily in the DARDAR satellite data. Of course, other ice containing clouds such as altocumulus or altostratus clouds may act as seeding clouds as well and may be the subject of a further study.*

AR: *In addition, we have changed the title to "How frequent is natural cloud seeding from ice cloud layers ($< -35\,°C$) over Switzerland?" to reflect the focus on seeding from ice cloud layers.*

RC: **Then I struggled with this classification: in situ origin cirrus ... and liquid origin cirrus! In former times, there was a clear separation between outflow cirrus (convectively generated cirrus, as remnants of big cumulus towers ...) and synoptic cirrus (cirrus uncinus, cirrocumulus, cirrostratus). And it was clear that outflow cirrus must have quite different properties than cirrus that formed homogeneously or heterogeneously via deposition nucleation. 'In situ' is some kind of a property and used to describe in situ observations, in situ experiments, in situ instruments, especially to contrast them from remote observation and remote sensing instruments .... But what about 'in situ origin' cirrus? ... I know what you want to say, but is that a proper designation? Is that even correct English? Locally generated cirrus vs convectively generated cirrus would sound better. Liquid origin: 'liquid' is not precise ... could be even sulfate aerosol droplets or oil droplets. One should discuss this kind of designation in a broader way to corroborate that this kind of classification of cirrus is open for discussion.**

AR: *We follow the cirrus characterization into in-situ and liquid origin clouds in Luebke et al. (2013), Krämer et al. (2016), Luebke et al. (2016), Wernli et al. (2016), Gasparini et al. (2018), Wolf et al. (2018), and Wolf et al. (2019) and thus also use their terminology. For a juxtaposition to the cirrus characterization by the synpotic situation in which the cirrus form, please refer to RC 2.4.*

**2.1. P2**

RC: **It took me some time after reading all the complex aspects given in the introduction what the goal of the study is: We concentrate on the cirrus impact only! I would suggest to simply state what the seeder-feeder mechanism is (definition), what kind (branches) of seeder-feeder pathways exist, and that you want to concentrate on the one via pure cirrus ..., and why you restrict your study to this specific field (because only for this one can use DARDAR ..., if I understood correctly). That would be more simple and straight forward.**

AR: *We see the need for clarifying this in the introduction and believe that this is achieved with the changed title and the additional sentences added in respondence to your first comment.*

**2.2. P3, Figure 3**

RC: **Case (c), the right panel is confusing. The orange line indicates $-35\,°C$ (?) and then you have ice (100%) above the respective height (at temperatures below $-35\,°C$)? ... and liquid water droplets (100%) below this height? Exactly for all temperatures higher than $-35\,°C$? Is that realistic? Maybe in the case of a thunderstorm with 30m/s upwind. . ... that may be the case, i.e., only water below the orange line. But with slow updrafts and ice sedimentation the picture is certainly more complicated.**

AR:    *We believe your comment refers to Figure 1c on page 3. This graphic illustrates the classification applied to the DARDAR data. As you state, we view the cloud as ice at temperatures lower than $-35\,°\mathrm{C}$. Everything below the isotherm, i.e. at higher temperatures, is viewed as a feeder cloud. In this study, we do not assume anything about the state of the feeder cloud. As stated in the caption, it is termed mixed-phase but could be liquid or ice phase. To make that more clear, we also changed $\Delta z_{\mathrm{il}}$ to $\Delta z_{\mathrm{im}}$ in response to RC 1.1. Contrary to what you say, we do not suggest that all hydrometeors in the feeder cloud are liquid, as is explicitly shown in the sketches in the top row of Figure 1.*

**2.3.   P9, l 193**

RC:    ***Please explain in more detail: You found scences with cirrus and liquid-water cloud in 32 % out of all cases, and then, in 77% out of these 32% cases, a liquid cloud directly below the cirrus? . . .so that the seeder-feeder process can work?***

AR:    *We added a clarifying sentence to the first statement:*

> 31 % *of the measurements contain both a cirrus and a mixed-phase cloud simultaneously. This is the percentage of cases in which a seeding of the lower cloud by ice crystals falling from the ice cloud above is possible. Tailoring this result to the sedimentation of ice crystals from a cirrus cloud,  when only the measurements that detect a cirrus cloud  are taken into account, in 75 % of these measurements also a mixed-phase cloud below them is detected.*

*The 32 % refer to the overall frequency of this seeder-feeder situation. But how frequent do we have a second cloud underneath a cirrus cloud? In 77 % of the times that we see a cirrus cloud, we also see a second mixed-phase cloud underneath.*
*Please note that the percentages stated changed due to errors noted in RC 1.11.*

RC:    ***Question: How do you know that the liquid-water cloud is free of ice crystals? Because of the radar observations? Please explain!***

AR:    *We do not know that the potential feeder clouds are liquid. We state several times in the paper that these clouds are termed mixed-phase, but could also be liquid or ice phase. To make that more clear, we also changed $\Delta z_{\mathrm{il}}$ to $\Delta z_{\mathrm{im}}$ as suggested in RC 1.1.*

**2.4.   P13, l 270**

RC:    ***I am sure that there are papers from the 1980s-1990s distinguishing the microphysical properties of outflow and synoptic cirrus. Please check! Or did this kind of research started just a few years ago as your references indicate?***

AR:    *Cirrus clouds can be characterized either by the synoptic situation in which they form or by their origin. The latter characterization was proposed by Luebke et al. (2013), Krämer et al. (2016), Luebke et al. (2016), Wernli et al. (2016), Gasparini et al. (2018), Wolf et al. (2018), and Wolf et al. (2019). As described by Wolf et al. (2019), "the same weather condition can contain both cloud types" (see also Krämer et al. (2020)), which leads them to conclude that "an origin-based parameterization of cirrus clouds seems to be more adequate and distinct in comparison to a weather-based parameterization". In line with the literature cited above we employ the characterization by origin in our investigation of the cloud microphysical properties. A comparison of the two characterization methods is beyond the scope of this article, especially because the*

*synoptic situation that the cirrus formed in is not examined.*

**2.5.  P13, l 273**

**RC:** ***By listing all available mechanisms, step by step, starting from temperatures above*** $-35\,°C$***, and then going to temperatures below*** $-35\,°C$ ***at which both homogenous and heterogenbous ice nucleation can occur. . .. the separation into different cirrus classes would be process-based and more easy to understand. Why do you not mention the immersion freezing process?***

AR:  *We agree, this was a long and compliated sentence. We rephrased it to make the classification more clear:*

> * Liquid origin clouds form from supercooled water droplets which are uplifted to the cirrus temperature range. They freeze either heterogeneously at warmer temperatures or predominantly homogeneously at temperatures  around* $-35\,°C$*. In the cirrus temperature range, cirrus clouds can also form by homogeneous nucleation of solution droplets or heterogeneous nucleation of ice nucleating particles. These cirrus clouds are termed in situ origin cirrus clouds.*

*Immersion freezing is not mentioned explicitly, because it is implicitly included as one type of heterogeneous ice nculeation through ice nucleating particles.*

**2.6.  P13, L285**

**RC:** ***What do you want to say?  : It also confirms . . .  that liquid origin cirrus clouds are composed of ice crystals. . . .I mean: a white horse is white. . . otherwise it is not a white horse.***

AR:  *A word is missing here. The sentence should read:*

> *It also confirms the finding from Luebke et al. (2016) that liquid origin cirrus clouds are composed of larger ice crystals.*

**2.7.  P13**

**RC:** ***I find the full discussion on pager 13 quite a bit too complex and many times confusing.***

AR:  *We rephrased these paragraphs and hope that in addition to the changes applied in response to your RC 2.5 this makes the discussion more clear.*

**2.8.  P18**

**RC:** ***My question remains:  Would be nice to have some speculation (some estimation, your opinion) on the relative impact of stratiform clouds (mixed-phase altocumulus etc . . . ) on the seeder-feeder processes.***

AR:  *As stated in response to your first comment, seeding ice crystals can also sediment from mixed-phase clouds. We state this in the Conclusion, and now also in the Introduction. We agree that it would be intersting to investigate natural cloud seeding from mixed-phase clouds, but this requires a reliable cloud phase satellite*

*product.*

**3. Reviewer Comment #3**

RC: *In their paper "How frequent is natural cloud seeding over Switzerland", the authors quantify the occurrence of an ice cloud layer above other clouds from CALIOP/CloudSat (DARDAR) data. In addition, the possibility of ice crystals sedimenting into the lower layers is calculated from temperature and relative humidity profiles based on ERA5. The paper is well written, the method is sound, the analysis is careful and the results are very interesting. I recommend the manuscript for publication after minor revisions.*

AR: *Thank you for your clear and thoughtful feedback. Please find our respective answers directly below your comments below.*

**3.1. Main comments**

RC: *I don't understand the "filtering" or smoothing of the data. Line 121 mentions a 7x7x7 points cube, but what are these three dimensions? Why is the median used? How sensitive are the results to this filtering?*

AR: *This is a mistake, the cube has only two dimensions since the filtering is applied before the DARDAR data is remapped. These are the horizontal dimension and the altitude, as we have know specified in the text:*

> *This cloud mask was found to be noisy and was therefore filtered (using a median filter over the surrounding 7 × 7 points plane, in altitude and horizontally along the track).*

*The cloud mask has values of 1 or 0 (cloud or no cloud). Using the median over the plane or the mean with threshold 0.5 to decide between 1 or 0 in a pixel is therefore the same thing, and using the median is simpler. Figure 1 illustrates the filtering results. We have evaluated other cube sizes as well and 7 seemed to be the most suitable size to exclude small holes in the clouds as well as small cloudy pixels from the analysis. Figure 2 shows the DARDAR results with and without filtering from tests we made in a previous version of the analysis.*

RC: *Line 131 says that reff is also filtered "for consistency", but the smoothing applied here is very different. What impact does this have on the results?*

AR: *The reason for why the two smoothing algorithms differ is a physical one: for the cloud mask we want to fill holes. After having done that, the smoothed field may see a cloud where the DARDAR product says that there is none. Now if we take the effective ice crystal radius from that pixel, it will give an unphysical value as it is taken from the product that sees no cloud. That is why we chose to use the median effective radius of clouds detected in the DARDAR data, in vicinity to the pixel in question, instead. We chose to only probe the vertical vicinity of the pixel in order to probe the real cloud base of the ice crystals. Applying no filter or the same filter as used for the cloud mask to the effective radii, their values were reduced compared to those derived using the vertical filter (see Figure 3).*

RC: *The temperature in the DARDAR dataset is also not a retrieved parameter, but is obtained from interpolated NWP data, if I'm not mistaken. How different is this from the ERA5 temperature? Is the discrepancy in the sublimation calculations only due to a discrepancy in relative humidity or also in temperature? How is this affected by the smoothing?*

[Figure]

Figure 1: Illustration of the cloud mask applied in the DARDAR analysis: cloud mask without (top) and with a $7 \times 7$ filter (bottom) for one exemplary satellite track.

[Figure]

Figure 2: Difference between DARDAR data with (green and gray) and without filtering (blue) from a previous version of the analysis: **(a)** Occurrence frequency of seeder-feeder situations (SF sit.) with respective $\Delta z_{\mathrm{im}}$ as a fraction of measurements (dark green/blue) or cirrus cloud measurements (light green/blue). **(b)** Cumulative occurrence frequency. For $\Delta z_{\mathrm{im}}$ a vertical resolution of $60\,\mathrm{m}$ is used. For comparison, the fraction of measurements with at least one cirrus cloud (light grey/blue) is given. Data from all tracks in the study time (2006 to 2017) and within the study domain were used. The shaded areas visualize the standard deviation of interanual variablility. This Figure corresponds to Fig. 3 in the manuscript.

[Figure]

(a) No filter for *reff*

(b) 2 dimensional cloud mask filter for *reff*

(c) 1 dimensional, vertical filter for *reff*

Figure 3: Distribution of *reff* for three different filter variants: **(a)** no filter, **(b)** $7 \times 7$ plane filter as described for cloud mask in the manuscript, **(c)** vertical filter as described for *reff* in the manuscript. The filter used for the cloud mask is the same in all three plots, i.e. the one described in the manuscript. Note that this Figure was produced with lower resolution data from a previous version of the analysis and therefore **(a)** differs from Fig. 4 in the manuscript.

AR:	*Indeed, the temperature in the DARDAR dataset is obtained from interpolated NWP data from ECMWF (Delanoë and Hogan, 2010a). As we state in the manuscript, we have compared some temperature profiles in the DARDAR and ERA5 data directly and have found up to $5\,°\mathrm{C}$ difference in the temperature. We used the relative humidity data from ERA5 in this study, and the DARDAR data does not contain the humidities used to create it, so we were unable to compare them directly. Of course, any difference in clouds and humidity between the reality that the satellites probe and the simulation in ERA5 is a source of error for the sublimation calculations, in the form of temperature as well as in humidity estimates. The smoothing discussed earlier is not applied to the environmental parameters. Therefore we do not expect this to contribute to the difference between the DARDAR and ERA5 datasets.*

RC:	**Sometimes the terminology is a little unconventional. The term "seeder-feeder" is also used for cases in which ice crystals fall into pure ice clouds. I don't think the term is appropriate in this case. Further, the ice cloud layer below $-35\,°\mathrm{C}$ is termed "cirrus", even if it is the upper part of a mixed phase cloud e. g. a frontal cloud or a convective cloud. I would reserve "cirrus" for isolated ice clouds.**

AR:	*Since we do not determine the phase of the lower lying clouds, we cannot exclude them from our "seeder-feeder" terminology. Similarly, since we do not separate into separate ice clouds and those that are part of a mixed-phase cloud, we chose to use one term for both ice cloud types. To avoid confusion we added a clarifying sentence in the Methods:*

> *Note that clouds termed mixed-phase could in principle be in the liquid or ice phase in reality, depending on their history and the presence of ice nucleating particles (see Fig. 1a). Similarly, in this study we denote all ice clouds at temperatures colder than $-35\,°\mathrm{C}$ as cirrus clouds, which could be isolated ice clouds or the upper parts of mixed-phase clouds.*

**3.2. Table 2**

RC:	*Table 2 gives coefficients "for cloud droplets", shouldn't this be ice particles?*

AR:	*The naming is correct. The parameterization for the velocity-mass-relation for cloud droplets from Seifert and Beheng (2006) is used for spherical ice crystals in this study. We think that this is appropriate because of the similar shape of the two hydrometeors. Due to this assumption we are overestimating the velocity of the spherical ice crystals, because a spherical ice crystal with the same mass as a liquid drop experiences a larger drag force acting on it. However, the relative overestimation for using the cloud droplet formulation for the spherical ice crystal is only about 3%, as detailed in the following for Stokes terminal velocity. With*

$m_i = m_l$, $r_i = \left(\frac{\rho_l}{\rho_i}\right)^{(1/3)} r_l$:

$$v_{T,l} = \frac{2}{9} \frac{r_l^2 g \rho_l}{\mu} \tag{1}$$

$$v_{T,i} = \frac{2}{9} \frac{\left(\frac{\rho_l}{\rho_i}\right)^{(2/3)} g \rho_i r_l^2}{\mu} \tag{2}$$

$$= \left(\frac{\rho_l}{\rho_i}\right)^{(2/3)} \frac{\rho_i}{\rho_l} v_{T,l} \tag{3}$$

$$= \left(\frac{0.92}{1}\right)^{(1/3)} v_{T,l} \tag{4}$$

$$= 0.97 \cdot v_{T,l} \tag{5}$$

*We added a clarifying statement to the caption of Table 2.*

**3.3. Figure 3b**

**RC:** *Why does the cumulative occurrence frequency for the situation with a cirrus cloud does not reach 1 minus the cirrus cloud frequency (i.e. a little less than 0.6) at 10km?*

AR: *What is shown in light green in Figure 3b is the fraction of cases with a potential feeder cloud within the measurements that contain a cirrus cloud. Thus, theoretically, if every situation with a cirrus cloud had another cloud below, it could go up to 100 %. In total, the fraction of measurements with a cirrus cloud is 42 % of all measurements.*

**RC:** *What is the sensitivity to the time step of the sublimation calculations?*

AR: *In tests performed with a more simple set up (constant temperature and relative humidity), the sublimation height of $50\,\mu m$ ice crystals was found to be independent of the time step, at least for those time steps $< 10\,s$, where there are no numerical instabilities (see Figure 4).*

**3.4. Line 140**

**RC:** *"mean values" of what?*

AR: *Both Vassel (2018) and Vassel et al. (2019) use "the average conditions of temperature, pressure and moisture of the individual subsaturated layers, measured by the radiosounding" for their sublimation calculations (Vassel, 2018). We corrected and further specified this in the manuscript as follows:*

> *For these parameters, Hall and Pruppacher (1976) used the NACA standard profile, while Vassel (2018) used mean values, and Vassel et al. (2019) used radiosonde profiles that were averaged for each subsaturated layer in their calculations.*

**3.5. Line 416**

**RC:** *I suggest to include the reference to the companion paper only if it is already available (at least as preprint) when the revised version of this manuscript is published. Otherwise, this is more confusing than helpful for the readers.*

[Figure]

Figure 4: Results of sublimation calculations testing the influence of the time step length (dt). The sublimation calculations were performed for a $50\,\mu\mathrm{m}$ spherical ice crystal and deviate from the ones employed in the manuscript only by using a constant temperature and humidity. The ice crystals start sedimenting from an altitude of $3000\,\mathrm{m}$ and the fall distance is the altitude at which the ice crystals sublimate. The calculations and this visualisation are based on Vassel (2018).

AR:    *We agree and removed the reference.*

**3.6.  Technical comments**

RC:    *Caption of Fig. 3: typo "atmposphere"*
       *Line 352: "for spheres": blanks missing.*
       *Table A2: typo "Earh".*

AR:    *Thanks for pointing out these typos. We corrected them in the new version of the manuscript.*

This document was generated with a layout template provided by Martin Schrön (`github.com/mschroen/review_response_letter`).

[revised manuscript text omitted]

 We split the dataset into one part with  $\Delta z_{im} > 100\,\text{m}$ and one with  $\Delta z_{im} < 100\,\text{m}$ as a proxy for the two cloud origins: in situ origin cirrus have large distances to the next underlying mixed-phase cloud, while liquid origin cirrus appear close to the $-35\,^\circ\text{C}$ isotherm. This separation indeed produces two different modes, as can be seen in Fig. 4b

300 and 4c. Figure 4b corresponds to liquid origin cirrus clouds. It displays larger ice crystals, from $\approx 35\,\mu\text{m}$ to $\approx 90\,\mu\text{m}$ at cirrus cloud base heights from $4500\,\text{m}$ to $9500\,\text{
[revised manuscript text omitted]